# Contrasting Forest Loss and Gain Patterns in Subtropical China Detected Using an Integrated LandTrendr and Machine-Learning Method



**Jianing Shen** [1,2], **Guangsheng Chen** [1,2,*] , **Jianwen Hua** [3], **Sha Huang** [4] **and Jiangming Ma** [5]

1   State Key Laboratory of Subtropical Silviculture, Zhejiang A&F University, Hangzhou 311300, China;
    jianingshen@stu.zafu.edu.cn
2   College of Environmental and Resource Sciences, Zhejiang A&F University, Hangzhou 311300, China
3   The Center for Ecological Forestry Development of Jingning County, Lishui 323599, China;
    evilive@stu.zafu.edu.cn
4   The Bureau of Agriculture and Rural Affairs of Lin'An District, Hangzhou 311300, China;
    2020602041069@stu.zafu.edu.cn
5   Guangxi Key Laboratory of Landscape Resources Conservation and Sustainable Utilization in Lijiang River
    Basin, Guangxi Normal University, Guilin 541006, China; mjming03@gxnu.edu.cn
*   Correspondence: chengu1@zafu.edu.cn

**Abstract:** China has implemented a series of forestry law, policies, regulations, and afforestation projects since the 1970s. However, their impacts on the spatial and temporal patterns of forests have not been fully assessed yet. The lack of an accurate, high-resolution, and long-term forest disturbance and recovery dataset has impeded this assessment. Here we improved the forest loss and gain detections by integrating the LandTrendr change detection algorithm with the Random Forest (RF) machine-learning method and applied it to assess forest loss and gain patterns in the Zhejiang, Jiangxi, and Guangxi Provinces of the subtropical vegetation in China. The accuracy evaluation indicated that our approach can adequately detect the spatial and temporal distribution patterns in forest gain and loss, with an overall accuracy of 93% and the Kappa coefficient of 0.89. The forest loss area was $8.30 \times 10^4$ km$^2$ in the Zhejiang, Jiangxi, and Guangxi Provinces during 1986–2019, accounting for 43.52% of total forest area in 1986, while the forest gain area was $20.25 \times 10^4$ km$^2$, accounting for 106.19% of total forest area in 1986. Although the interannual variation patterns were similar among three provinces, the forest loss and gain area and the magnitude of change trends were significantly different. Guangxi has the largest forest loss and gain area and increasing trends, followed by Jiangxi, and the least in Zhejiang. The variations in annual forest loss and gain area can be mostly explained by the timelines of major forestry policies and regulations. Our study would provide an applicable method and data for assessing the impacts of forest disturbance events and forestry policies and regulations on the spatial and temporal patterns of forest loss and gain in China, and further contributing to regional and national forest carbon and greenhouse gases budget estimations.

**Keywords:** forest gain; forest loss; LandTrendr; Random Forest (RF); forest policies; subtropical China

## 1. Introduction

The earth is becoming greener, and China is regarded as the major contributor to this trend [1,2]. According to the 1st and 9th National Forest Inventory (NFI), forest coverage in China has increased from 12.7% in 1976 to 22.96% in 2018. Unlike other countries, increasing forest resources in China were primarily owed to the issuance and adjustment of national forestry policies [3–5]. During the past half century, China has put forward a series of forestry policies and regulations, such as timberland base shifting, "grain to green" program (GTGP), and Natural Forest Conservation Program (NFCP). In addition, many regional and national afforestation projects have been implemented, such as the Three-North Shelterbelt

Project in northern, northeastern, and northwestern China, Project for Fast-growing and High-yield Plantation in the Key Areas (FHPKA), and Shelterbelts Project along the Upper and Middle Reaches of the Yangtze River [3,6–8]. The implementations of these forestry policies and projects had not only greatly improved forest quality and quantity, but also significantly changed the structure and function of forest ecosystems in China [1,3,9–12]. Meanwhile, forest losses in China were also increasing due to forestry policies and intensive disturbance events such as land use change, insects and diseases, fire, strong winds, and harvesting [12]. Although the total forest area was increasing, the natural forest area was decreasing and forest age became younger under both the influences of both forestry policies and disturbances. For example, the "three determinations" policy (i.e., determining the rights for forest land property, privately planted forests, and forestry production) in 1981 [13] and the collective forest tenure reform (CFTR) policy in 2003 have caused sudden increases in forest harvesting and area losses [5,14,15]. Diseases outbreaks have affected 22,954 km$^2$ of forest area in 2019, and insects and pests have affected 81,146 km$^2$ of forest area according to the NFI report. In addition to forestry policies and disturbances, other factors such as elevation, local economic development condition, and local forestry policies were reported to affect forest gain and loss [16–18]. All above factors could greatly affect the spatial and temporal patterns in forest gain and loss at different scales; however, few regional studies have assessed their impacts in China [19]. The lack of a spatiotemporally explicit and high resolution forest gain and loss dataset is the main impediment for filling this knowledge gap. Forest loss and gain cannot be fully reflected by many existing land use and cover datasets [19–21], which can only reflect the long-term land cover changes from or to forests but not be able to track the short-term changes in forest status due to scattered and short-lasting disturbance events.

With the rapid progress in remote sensing detection technology and increasing publically available satellite images, the remote sensing approach has been widely applied to detect forest gain and loss at local, regional, national, and global scales [22–25]. In these studies, many satellite platforms were applied, such as AVHRR, MODIS, Landsat, SPOT, Sentinel, and Worldview. Due to its long-term history and relatively high spatial and temporal resolution, Landsat imagery was the most popular platform used for detecting forest loss and gain at regional, national, and global scales [24,26–29]. The early detection methods of forest loss and gain were mainly based on dual-temporal images, which mainly obtained the change area by comparing two period classified images, or computing the image difference, ratio, and canonical correlation changes. More recently, the time series analysis method has been widely applied to identify forest disturbance and recovery through background noise and time-series trajectories of abrupt changes. Many change detection algorithms were developed based on the time series satellite images, such as BFAST (Breaks for Additive Season and Trend Monitor) [30], LandTrendr (Landsat-based detection of Trends in Disturbance and Recovery) [31], VCT (Vegetation Change Tracker) [27], and CCDC (Continuous Change Detection and Classification) [32]. Different wavelength bands, vegetation indices, image conversion processing, and sub-pixel variable ensembles were applied in these change detection algorithms, which greatly improved the automation and accuracy of forest loss and gain detection. Among these algorithms, LandTrendr solely requires one or two years to input a cloudless image or multiple cloud-covered images, and may not only capture the long-term slow forest changes but also detect the mutation change trend, and thus gains a better applicability [22,31,33]. At the early stage, LandTrendr only used the Normalized Burn Ratio (NBR) vegetation index to detect the maximum change magnitude and thus determine the forest loss and gain [31,34]; however, it is likely that a single spectral index is insufficient to adequately characterize forest loss and gain [35]. Cohen et al. [18,35] demonstrated that the use of a single algorithm with multiple indices to employ ensemble stacking can improve the disturbance detection. They identified six most important factors including NBR, reflectance of green band (Band2), short-wavelength infrared band 1 (Band5), Tasseled Cap index of greenness (TCG), brightness (TCB), and wetness (TCW) for the ensemble stacking. Based on these input variables, they applied the

Random Forest (RF) machine-learning method for the secondary classification of forest loss and gain. In addition, several other studies have also proved the better performance of this integrated approach. For example, [36] proposed an ensemble approach for predicting disturbance and subsequent recovery levels for each disturbed pixel, and argued that this approach could simultaneously describe the levels of both disturbance and subsequent recovery as a single hybrid pattern; [37] developed the integrated approach between the LandTrendr and gradient nearest neighbor (GNN) imputation method to detect large tree and snag distributions; [38] detected forest harvest and post-harvest recovery in Japan using the integrated RF and LandTrendr approach, and concluded that this approach can provide valuable information on a country-wide replantation activity in harvest areas. However, due to the more complex terrain, forest types, and climate conditions in China, our previous study [16] indicated that some other factors such as elevation and image textures could also be important factors for detecting forest loss and gain in China.

At present, there are only a few studies in China that have assessed the spatial and temporal patterns of forest loss and gain on a small scale, such as mountains, counties, and multi-counties [17,39–42]. Most of these studies directly adopted the forest change detection algorithms in China and the parameters and methods were not localized and improved. At present, the Global Forest Change (GFC) product including both forest loss and gain in 2001–2020 was available for providing forest loss and gain data covering the entirety of China; however, this product has not been specifically validated based on local forest state data [43]. Based on multiple data comparisons, several previous studies have also indicated that the GFC product significantly underestimated the forest loss and gain area at stand [44], national [19,45], and global scales [46]. In addition, this product only covered recent years and missed the 1980–2000 period when China has experienced stronger disturbance and initializations of several afforestation projects [16]. Therefore, it is necessary to localize and improve the forest change detection methods for China by considering the complicated forest and topographic conditions and training and evaluating the algorithms using more intensive local observational data, and generate provincial and national forest loss and gain dataset using the validated methods.

The forest coverage (>59%) in the Jiangxi, Guangxi, and Zhejiang Provinces ranks in the top five highest provinces in China and thus have more forest resources compared with other provinces in China. Although all three provinces were influenced by similar national forestry laws, policies, and regulations, the forestry productions in these three provinces have witnessed very different trajectories due to the large differences in their execution capability of the national forestry laws, policies, regulations, economic development levels, population growth rates, provincial forestry policies, and topographic conditions. The spatiotemporal patterns in forest cover and the impacts of these socioeconomic factors are still unclear. Based on all available Landsat time-series images and Google Earth Engine (GEE) platform, our study intended to recalibrate the LandTrendr change detection algorithm and identified the most important variables in RF classifier for their applications in subtropical China. Specifically, our objectives were to (1) detect the forest loss and gain in the Jiangxi, Guangxi, and Zhejiang Provinces of subtropical China using an integrative LandTrendr and RF method and further evaluate the performance of this approach; (2) analyze and compare the spatial and temporal patterns of forest loss and gain in the three provinces during 1986–2019; and (3) identify the impacts of major forestry policies on the temporal change patterns of forest loss and gain area. In this study, forest loss denotes the stand-replacing disturbance (tree loss fraction > 70%), including both permanent loss due to land conversion and temporary loss followed by tree regeneration. Forest gain denotes the inverse of loss, or the establishment of tree canopy from a non-forest state [23]. Our study establishes a well-evaluated integrative approach for detecting forest loss and gain in China, and provides accurate data for the establishment of national level forest loss and gain dataset, reconstruction of forest age structure, and large-scale modeling of forest carbon stocks and fluxes. In addition, our study results can also provide an accurate assessment on the effectiveness of major forestry policies and regulations since the 1980s.

## 2. Materials and Methods

### 2.1. Study Region

The study region is located in the three provinces of subtropical China, including Zhejiang, Jiangxi, and Guangxi Provinces (Figure 1). All three provinces are located in the subtropical monsoon climate region. Zhejiang Province (118°01′E–123°10′E, 27°02′N–31°11′N) is located on the southeastern coast of China. This province has a complex terrain, with mountainous/hilly and plain terrains accounting for 70% and 23%, respectively. The average annual temperature is about 18 °C and annual precipitation ranges between 1000 and 1900 mm. The mean annual sunshine hours range between 1710 and 2100 h. Jiangxi (113°34′E–118°28′E, 24°29′N–30°04′N) is located in the center of southeastern China. Mountainous, hilly, and plain area accounts for about 36%, 42%, and 12% of the terrains, respectively. The average annual temperature ranges between 16 and 20 °C and annual precipitation ranges between 1300 and 2000 mm. The Guangxi Province (104°26′E–112°04′E, 20°54′N–26°24′N) is located in the southern coast. Mountainous, hilly, and plain area accounts for about 57.8%, 28.9%, and 6.3% of the terrains, respectively. The province is mostly karst landform, accounting for about 38% of the land area. Most areas of Guangxi have a subtropical monsoon climate, and a few areas are affected by the tropical monsoon climate and are rich in heat. The average annual temperature ranges between 17 and 23 °C and annual precipitation ranges between 1400 and 2100 mm, with most (>70) rainfall occurring in summer. According to the 9th NFI report (2014–2018) (http://forest.ckcest.cn/sd/si/zgslzy.html; accessed on 15 March 2022), the forest coverage rates in Zhejiang, Jiangxi, and Guangxi are 59.43%, 61.16%, and 60.17%, respectively, ranking as the fifth, second, and fourth, respectively, among all China's provinces. The main tree species are subtropical evergreen broad-leaved forest and evergreen needleleaf forest.

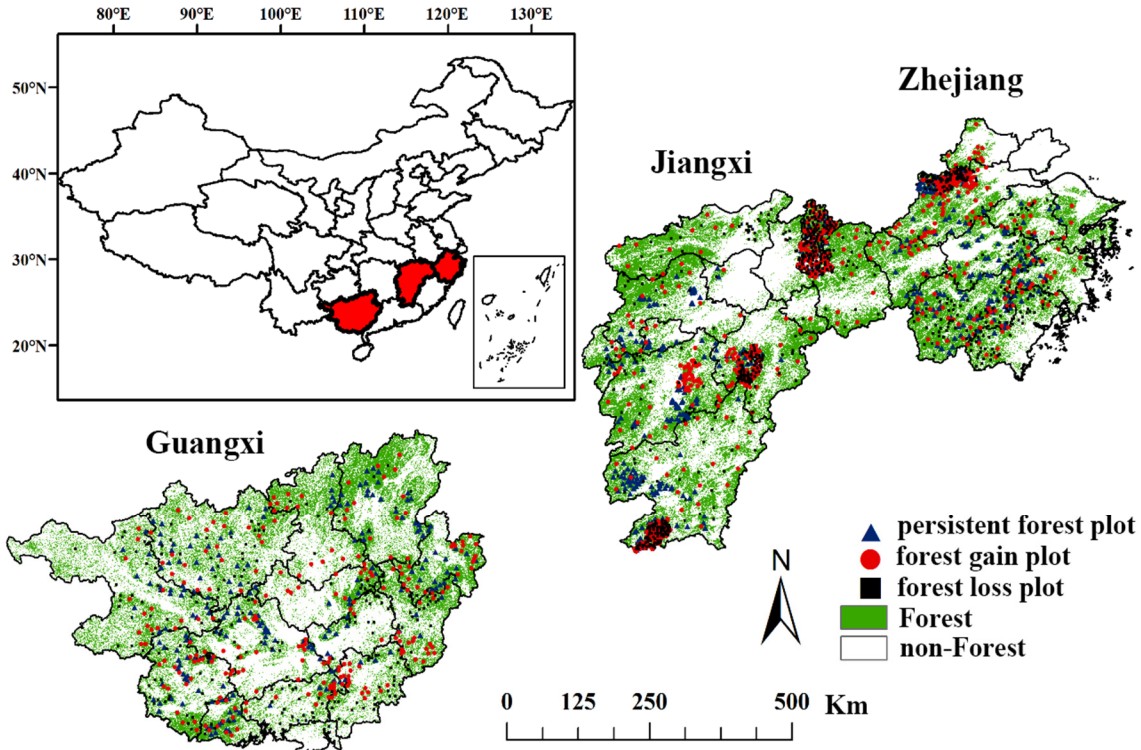

**Figure 1.** The location of study region and the spatial distribution of different types of sampling plots.

Guangxi Province is oriented as the new timberland base for China under the support of "Project for Fast-growing and High-yield Plantation in Key Areas" (FHPKA), one of the ten biggest forestry projects in China. The Guangxi forest land area only accounts for about 5% of the national total forest area, but it produced about 50% of the national total

wood products in 2020. The main fast-growing and high-yield tree species is eucalyptus (*Eucalyptus robusta Smith*), which has a short rotation age of around 6–8 years and accounted for about 70% of the total wood production in Guangxi in 2017. The gross forestry output value has increased from 1.81 billion RMB in 1990 to 43.74 billion RMB in 2020, with 24 times of increases. The gross forestry output value in Zhejiang Province has increased from 1.60 billion RMB in 1990 to 18.96 billion RMB in 2020, with 12 times of increases. The policy for public welfare forest classification and conservation in Zhejiang started in 2001, which is the earliest province in China. The forestry production in Zhejiang mainly targeted to realize ecological forestry, high-income forestry, and humanity forestry. Non-destructive and sustainable forest resource usage became the top priority, and thus the forestry output value was mainly from the bamboo forests and high-profit economic forests. Jiangxi Province was in the transitional stage from forestry-based economy to agriculture-based economy. The gross forestry output value has increased from 2.40 billion RMB in 1990 to 36.79 billion RMB in 2020, with 15 times of increases.

*2.2. Data Sources and Descriptions*
2.2.1. Satellite Data and Preprocessing

The acquisition and processing of remote sensing images and the operation of algorithms were conducted on the GEE platform (https://earthengine.google.com/; accessed on 1 February 2022). GEE is a PB-scale scientific analysis and geospatial data set visualization platform, including Landsat, MODIS, Sentinel, and other massive geographic and remote sensing data, and has powerful data storage and computing capabilities [47]. In order to reduce the influence of atmospheric solid or liquid aerosols scattering and Rayleigh scattering, the SR data of the ground surface reflectance that have been corrected for atmospheric and radiation are selected [48,49]. We collected the tiles with IDs: "LANDSAT/LT05/C01/T1_SR", "LANDSAT/LE07/C01/T1_SR", "LANDSAT/LC08/C01/T1_SR", and "USGS/SRTMGL1_003", which were corresponding to the Landsat5 TM, Landsat7 ETM+, Landsat8 OLI, and the 30-meter resolution SRTM DEM data [49,50]. The surface reflectivity of Landsat5 TM and Landsat7 ETM+ was obtained by radiation calibration and atmospheric correction using the LEDAPS model algorithm [49]. In order to reduce the interference caused by phenological changes while ensuring the image quality requirements, Landsat images of the growing seasons from May to October each year were selected to construct the Landsat time series stack (LTSS). In order to reduce the error caused by the sensors during change detection and construct Landsat time series data with more consistent radiation characteristics, this study used the conversion coefficient from Roy et al. [51] to process the Landsat8 OLI data. The preprocessing such as cloud masking, shadow removal, stitching, and cropping were already included in the GEE LandTrendr program package [52]. Finally, the numbers for available images from different Landsat platforms in each year were shown in Figure 2.

2.2.2. Training and Validation Sampling Data

A large number of sample point data were collected for the training of the RF classifier and the validation of the classification results. The sample plots came from three sources, including local forest management inventory plots, NFI (National Forest Inventory) plots, and the visual interpretation plots based on high resolution satellite images from Google Earth Pro platform. A total of 2516 sample points were obtained, including 884 forest recovery sample points, 967 forest loss sample points, and the rest were persistent forest sample points (Table 1). Based on the high resolution satellite images (spatial resolution < 5 m; mostly Quickbird imagery), the 10 m Sentinel-2 panchromatic images, and 30 m Landsat images on Google Earth Pro, we visually interpreted the forest loss and gain plots. We first identified all available high-resolution images during the 2010–2019 period, and then compared the images at two time periods (<3 years interval) to identify the forest loss and gain area and occurrence time (year). During the visual interpretation, three adjacent sample periods were selected (i.e., pre-disturbance, post-disturbance, and disturbance year).

The high-resolution images were not always available at the <3 years interval; therefore, to more accurately determine the actual loss and gain years, we also loaded the available 10 m Sentinel-2 panchromatic images (only for the period after 2016) and 30 m Landsat images (true color composite band). Under the cases with few available high-resolution images (i.e., the span years for two high-resolution images are greater than 3 years), these Sentinel-2 and Landsat images will be used to assist the determination of forest loss and gain years. Use the Landsat images as a boundary map, we only selected sample plots at the area with forest loss or gain greater than 30 m × 30 m, and with loss or gain fraction greater than 70% at a 30 m × 30 m pixel level. We classified three types of forest status: the forest loss (deforestation), forest gain (afforestation or reforestation), and persistent (stable) forest. Totally, 1258 sampling plots were visually interpreted, among which 472 plots for forest loss, 513 plots for forest gain, and 273 plots for persistent forest (Table 1).

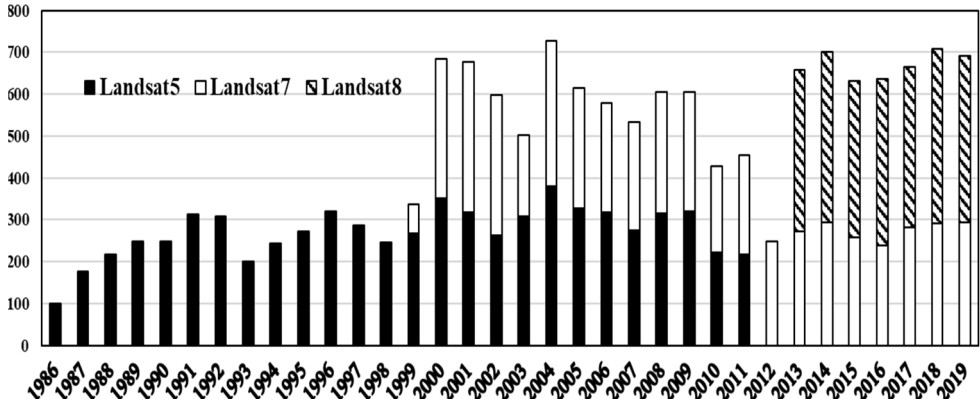

**Figure 2.** Available image tiles from different Landsat platforms during the growing seasons of 1986–2019 in the study region.

**Table 1.** The sources, categories and numbers of sampling plot data. NFI: national forest resource inventory; LFMI: local forest management inventory.

| Categories | FGV | FGT | FLV | FLT | PFV | PFT | Total |
|---|---|---|---|---|---|---|---|
| NFI | 13 | 46 | 20 | 42 | 131 | 79 | 331 |
| LFMI (Linan) | 23 | 65 | 24 | 78 | 15 | 52 | 257 |
| LFMI (Jingdezhen) | 28 | 83 | 37 | 111 | 19 | 17 | 295 |
| LFMI (Longnan) | 19 | 59 | 18 | 57 | 17 | 18 | 188 |
| LFMI (Yihuang) | 20 | 56 | 18 | 49 | 21 | 23 | 187 |
| Visual interpretation | 213 | 259 | 220 | 293 | 117 | 156 | 1258 |
| Total | 316 | 568 | 337 | 630 | 320 | 345 | 2516 |

This study obtained the 6th (1999–2003) NFI plot data. The NFI plot size is about 800 m$^2$, which is close to a Landsat grid cell size (900 m$^2$). There were more than ten thousand NFI plots for the three provinces; however, most plots consisted of persistent forest area. Eventually, we collected 331 sample plots from NFI, among which were 59 plots for forest loss, 62 plots for forest gain, and 210 plots for persistent forest.

The Local Forest Management Inventory is usually carried out at the county level for the purpose of better organization and management of local forest resources. Compared with the NFI the basic unit of the Local Forest Management Inventory is the small class (forest stands with relatively consistent attributes), which can conduct a more in-depth investigation of the forest resources in the entire county. The accuracy of the recorded forest change information is sufficient for Random Forest training and result verification.

The spatial distribution training and validation plots were shown in Figure 1. All sample plots were further divided into 6 categories according to their usages: persistent forest training (PFT), persistent forest validation (PFV), forest loss training (FLT), forest

loss validation (FLV), forest gain training (FGT), and forest gain validation (FGV) (Table 1). The recorded information for all plots included: the coordinates, forest change category (gain, loss, or persistent forest), and the year of gain or loss. This information will be used to match the LandTrendr and RF classifier outputs.

### 2.2.3. Other Data Sources

This study collected the inventory data from the 9 times of NFI to compare our classified results. The NFI was conducted every five years, and hundreds of thousands of fixed sample plots were deployed across the country. The forest status (loss and gain) information was recorded for all forest plots. Based on these plot data, the area of forest loss due to different disturbances (i.e., fire, insect, disease, and harvesting) and gain (i.e., afforested area) at the provincial and national levels was extrapolated based on statistical methods and was available for downloading from the NFI website (http://forest.ckcest.cn/sd/si/zgslzy.html; accessed on 15 March 2022). In addition, the forestry output values and annual planted forest area in each province were also downloaded from the NFI website. These data generally covered the period from 1990 to 2018. The global forest change (GFC) data product (version 1.7) [23] was also downloaded from the GEE platform to compare our detected forest loss and gain at both spatial and temporal scales from 2001 to 2019. The tree canopy change data from Song et al. [53] were also downloaded and analyzed for comparisons of forest gain area.

There were over 289 forest laws, policies, and regulations that were put forward during 1949–2020 [6]. Among them, the major forestry laws, policies, and regulations that can affect forest loss and gain in the three provinces were selected and are described in Table 2.

### 2.3. Data Processing

The approach by integrating the LandTrendr algorithm with the RF method was applied to detect forest loss and gain. The main flow work is listed in Figure 3. Below, we briefly described the procedures for the integrated LandTrendr and RF approach. More detailed descriptions can be referred to our previous study [16].

### 2.3.1. Extraction of Forests

The non-forest land has spectral characteristics close to forests, and it is easy to be classified as forest after changes in some phenological or man-made images during forest change detection [23]. Therefore, exclusion of other land cover types could reduce some uncertainties for forest loss and gain detection. The RF classifier was applied for forest classification. The sixth NFI (1989–1993) plot data (2300 forest plots) were used as training samples to classify forest distribution in 1986, while the 665 training and validation sample plots (Table 1) were used as training to classify forest distribution in 2019. In order to make the extraction result of the forest more accurate, in addition to the band, we also use the vegetation index, texture information, and elevation as classification features. Combining the forest extraction results in 1986 and 2019, if both classifications are classified as non-forest, the raster will be eliminated. Mask the annual composite image with the last retained forest distribution map, then the remote sensing image containing only the forest area is obtained to participate in the subsequent interference detection and analysis calculation.

### 2.3.2. LandTrendr Algorithm and Implementation

The LandTrendr time series segmentation algorithm was considered to be one of the best algorithms for detecting both slow and abrupt changes of forests [31]. A set of spectrum-time segmentation algorithms in LandTrendr can effectively detect forest interference, and can be used to generate trajectory-based spectral time series data without interannual signal noise [31,34]. The algorithm can not only identify short-term severe disturbances but also long-term slow vegetation change events. The LandTrendr version on GEE was from Kennedy [52]. This version has simplified the Landsat image pre-processing steps, allowing

focuses on the translation of the core temporal segmentation algorithm. The implementation of LandTrendr's algorithm included the following steps [31,35,52] (Figure 3): (1) removal of the spikes (i.e., the noises caused by cloud, snow and shadows): A de-spiking algorithm was developed in LandTrendr to remove the abrupt noises. (2) Identify potential segmentation points (vertices): A regression-based vertex identification strategy is used to identify vertices. The excessive vertices are removed based on a criterion. (3) Fit the spectrum trajectory: After the segmentation points are determined, a series of fitting methods are used to determine the spectral index value of each segmentation point, thereby forming the best continuous spectral index trajectory in the entire time series. (4) Simplify the models and select the best model: Fitting the spectral trajectory obtains the most complicated segmentation model, so the model should be simplified to highlight the spectral trajectory characteristics and to filter the noises.

**Table 2.** The selected major forestry policies and regulations that affect forest loss and gain in the subtropical China.

| Law, Policies, Regulations | Timeline | Key Policies | Effects |
|---|---|---|---|
| Decision on Several Issues Concerning the Protection and Development of Forests | 1981 | The three determination policy | The private party have the right to own and manage forests |
| Forestry Law of the People's Republic of China | 1984 | The first forestry law in China | Constraints on forest activities |
| Guidelines on Enhancing the Management of Collective Forest Resources in the South and Prohibiting of the Indiscriminate Tree Felling | 1987 | The three determination policy was found not suitable and stopped; the regulations for forest cutting quota | Enhanced forest management and protection |
| Notice on Protecting Forest Resources and Stopping deforestation, Reclamation and Indiscriminate Occupation of Forest Land | 1998 | Trial time for the NFCP | Began to protect forest resources especially natural forests |
| Regulations on Returning Farmland to Forests | 2002 | The starting time of GTGP policy | Recover forest coverage |
| Project for Fast-growing and High-yield Plantation in Key Areas (FHPKA) | 2002 | The starting time for shifting timberland base to the south | To protect natural forest resource in northern China |
| Decision on Accelerating Forestry Development | 2003 | The trial time for the collective forest tenure reform (CFTR) | Separation of ownership, contracting right and management right of collective forest land |
| Opinions on Comprehensively Promoting the Reform of Collective Forest Right System | 2008 | Fully implementation of the CFTR | The CFTR policy was found effective and thus widely applied |
| Outline of National Forest Land Protection and Utilization Plan | 2010 | The starting time for second stage of the NFCP | Full implementation of the NFCP |
| Guidelines for National Public Welfare Forests Management | 2013 | The starting classification of public welfare forests | More strict conservation for restoring forest ecological function |

**Table 2.** *Cont.*

| Law, Policies, Regulations | Timeline | Key Policies | Effects |
| --- | --- | --- | --- |
| The Guidelines for the Reform of State-owned Farms | 2015 | Reform of state-owned forest farms (RSFF) | Forest farms shift from a profit-making agency to forest protection agency |
| Regulations on the Implementation of the Forestry Law of the PRC China (amendment 2016) | 2016 | Institutional guarantee for deepening forestry reform | Protect forest resource and realize ecological civilization |

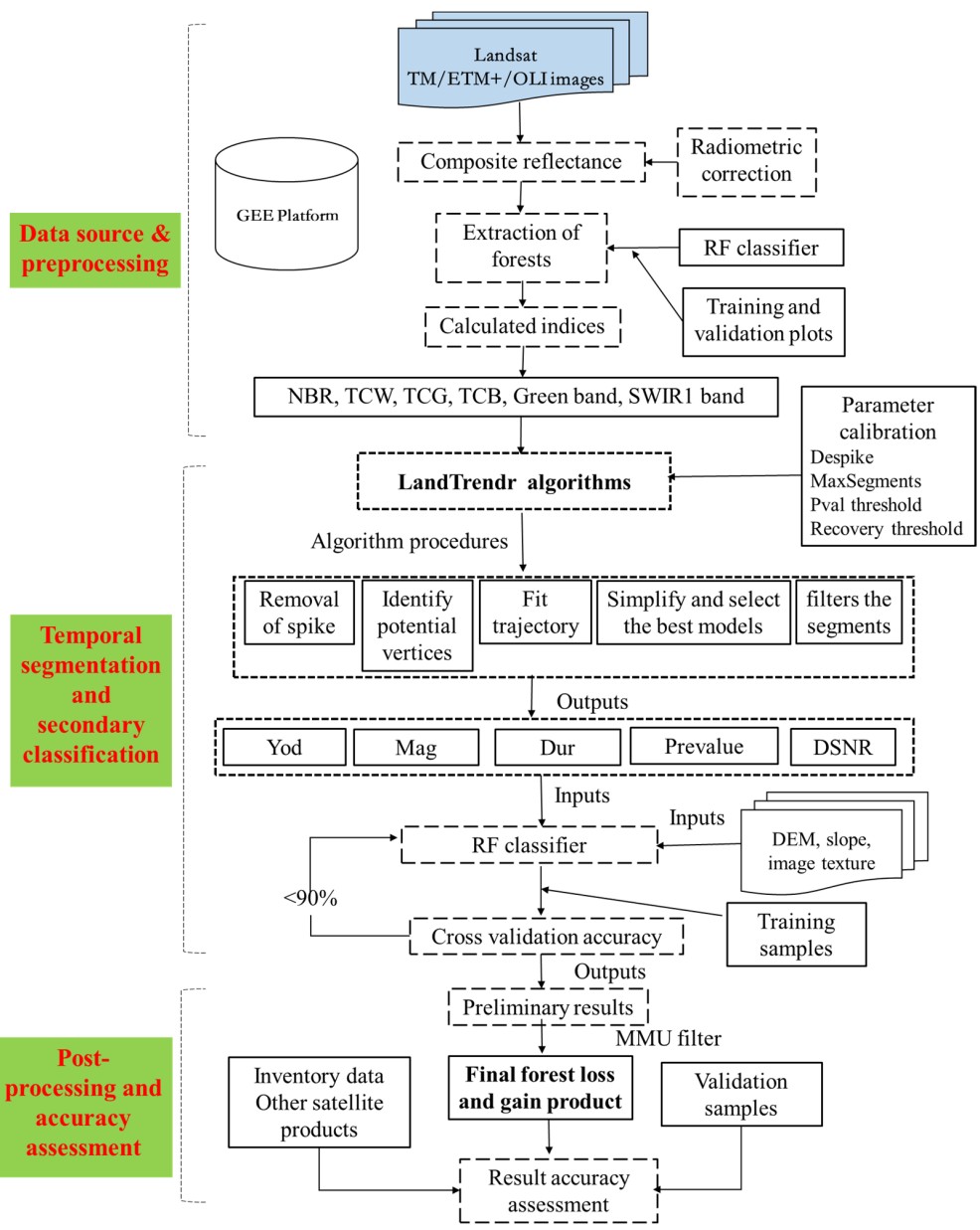

**Figure 3.** The work flow for forest loss and gain detections by integrating the LandTrendr algorithm with the Random Forest machine-learning method. Note: NBR, TCB, TCG, and TCW are vegetation indices. Green: green band (Band2); SWRI1: short-wavelength infrared band 1 (Band5); RF: Random Forest; Yod (year of detection), Mag (magnitude), Dur (Duration), Prevalue, and Dsnr (Detail Signal Noise Ratio) were the outputs from LandTrendr.

Regularly, the single NBR index was used to detect forest loss and gain in LandTrendr [31,34]. However, Cohen et al. [18,35] found that the single NBR could cause large errors due to the image quality; instead, the use of multiple indices to employing ensemble stacking can improve the disturbance detection. We chose the index combination with the lowest overall error as suggested by Cohen et al. [18,35] including TCB, TCG, TCW, NBR, green band reflectance (Band2), shortwave infrared band 1 (SWIR1 or Band5) to feed and run the LandTrendr algorithm. The parameters including max_segments (6), despike (0.8), pval (0.05), and recovery_threshold (0.5) for LandTrendr were tested and slightly modified based on the default values provided by Cohen et al. [18,35]. The output results were an array images with 5 outputs × N years. These output variables are yod (year of detection), mag (magnitude), dur (duration), preval (prevalue), and Dsnr (Detail Signal Noise Ratio). Each grid of the outputs includes a series of segments. We used five or four years as a time window to extract the maximum change magnitude (decrease or increase) within a window based on each of the 6 input variables. Then, we exported the magnitude data for each five year period (1986–1990, 1991–1995, 1996–2000, 2001–2005, 2006–2010, 2011–2015, 2016–2019). Due to the difficulty in detecting forest recovery in a shorter time span for the 2016–2019, the detected forest gain patterns in 2016–2019 were not analyzed in this study. In the ordinary LandTrendr algorithm, the forest gain and loss are identified based on the magnitude data for NBR [28,31], and the Yod with the largest segment, which represents the maximum change of NBR in a time window, is considered as the year with loss or gain. However, not all these identified largest segments are actually forest loss or gain. Some of the identified segments may be due to the issues of image quality or other noises. Other indices are necessary to be used as complementary evidence [35]. Therefore, all segmented outputs from 6 indices were used as input variables to further make decisions on forest loss and gain in the RF classifier.

### 2.3.3. Secondary Classification Using the RF Method

The RF classifier was further used to identify the final forest loss and gain based on the LandTrendr results. The LandTrendr obtained segmented outputs including mag, dur, preval, and dsnr for each index and totally 24 variables (4 outputs × 6 indices) were used as the inputs in the RF classifier. In this study, a 5-year segment window was applied, so the RF classifier was run every 5 years. Except for the outputs from LandTrendr, other assisting variables including 8 image texture variables, elevation and slope were also used as the input variables of RF. These assisting variables have also been used in previous studies to detect forest loss and gain [33,36]. The 1543 training samples (Table 1; Figure 1) were used as training for the decision making algorithm in the RF classifier. In the RF classifier, it randomly chooses 80% of the training points to train the algorithm and the remaining 20% points are used to cross-validate the fitted algorithm. If the cross-validation accuracy is higher than 90%, all training points are used again to obtain the final algorithm. Otherwise, the parameters for NumberOfTrees and VariablesPerSplit will be automatically adjusted until the accuracy meets the requirements. The RF outputs variables include forest loss, gain, persistent forest, and persistent non-forest. Due to the limitations of image quality and the LandTrendr algorithms, our detected forest loss and gain area only covered those with greater than 70% tree loss or gain within each pixel. In another word, our study can only accurately monitor the forest loss and gain due to stand-replacing events.

### 2.3.4. Post-Processing and Accuracy Assessment

Due to the likely mixture pixel issue, image quality, system errors, and other unidentified issues, the identified forest loss and gain pixels were widely scattered or fragmented. We applied the minimum mapping unit (MMU) filtering to remove these dispersed pixels [35]. The minimum unit is set as 4 pixels (0.36 ha) to perform the MMU filtering. The outputs after MMU filtering were the final products for forest loss and gain during 1986–2019.

Based on the 973 validation sample points (Table 1) and selected 351 pixels from GFC product, the classification accuracy was assessed. The validations were performed by separating the loss, gain, and persistent forest categories (Table 3). Specifically, for the validations using sample points, the producer and user accuracies for forest loss were about 94% and 95%, respectively; the producer and user accuracies for forest gain were 94% and 93%, respectively. The overall accuracy and Kappa Coefficient for both forest loss and gain were about 93% and 0.89, respectively. The detected results were further compared with the results from GFC product [23]. The comparison indicated that the overall accuracy and Kappa Coefficient for both forest loss and gain were 91% and 0.86, respectively, suggesting a well-match between the GFC product and our detected results. Overall the accuracy assessment results indicated that the integrated LandTrendr and RF method can adequately detect forest loss and gain. Our sampling strategy was based on an assumption that the plot assemblage is sufficiently similar to the population; however, this assumption cannot be fully tested by our sampling design. The sample plot numbers were limited and locations were based on the availability of high-resolution images (not completely random). Therefore, our inference for the accuracy assessment may be slightly biased.

**Table 3.** Accuracy assessment confusion matrix for forest loss and gain classification results. Note: the pixel values were compared for the GFC product.

| Data Sources | Change Types | Loss | Gain | Persistent | Produce Accuracy | User Accuracy | Overall Accuracy | Kappa Coefficient |
|---|---|---|---|---|---|---|---|---|
| Sampling points | Loss | 301 | 4 | 11 | 95% | 94% | | |
| | Gain | 6 | 314 | 17 | 93% | 94% | 93% | 0.89 |
| | Persistent | 12 | 15 | 293 | 92% | 91% | | |
| GFC product | Loss | 114 | 2 | 4 | 95% | 93% | | |
| | Gain | 3 | 100 | 8 | 91% | 89% | 91% | 0.86 |
| | Persistent | 5 | 12 | 103 | 86% | 90% | | |

We have evaluated the forest change detection mechanisms in LandTrendr in our previous study for Jiangxi Province alone [16]. Based on the three selected disturbed forest locations, here we further evaluated the change detection mechanism (Figure 4). Comparing with the high-resolution, Landsat images were not very clear and had some mixed pixels, but the forest loss area can still be easily and accurately delineated based on Landsat images, only showing some mismatches for the mixture pixels at the edge of forests. Compared with the GFC product, our detected forest loss area was more matched with the visually interpreted forest loss boundary at spatial scale. The fitting mechanisms based on NBR values were further evaluated at the selected sample plots. The results indicated the fitted disturbance trajectories from LandTrendr can successfully detect the forest loss in the years 2016, 2018, and 2019 at the forests in Ningming County, Yongfu County, and Rong County, Guangxi Province.

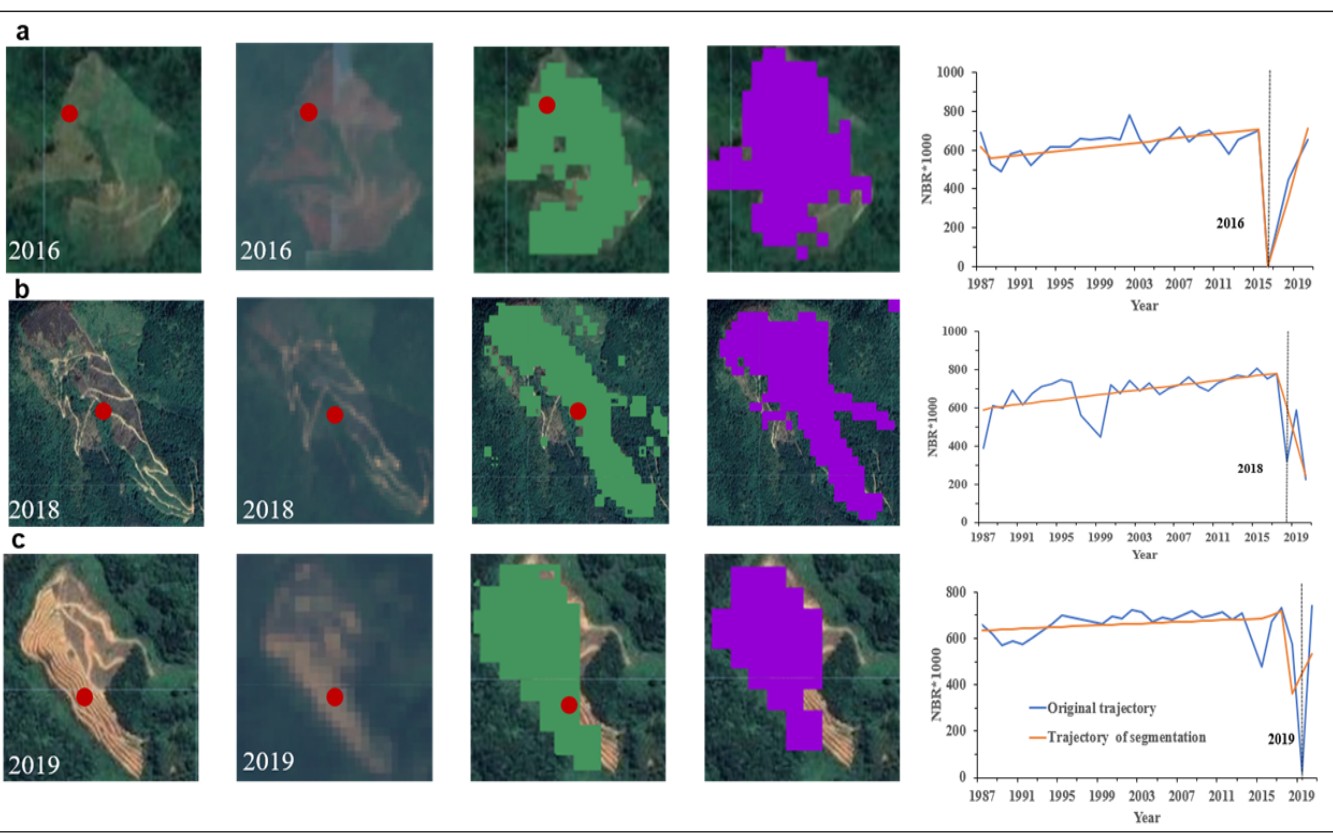

**Figure 4.** The comparisons of detected forest loss area with the GFC product at selected disturbance areas in (**a**) Ningming County, (**b**) Yongfu County, and (**c**) Rong County in Guangxi Province. Note: the red dots are the sampling plots, which are also the points for fitting the disturbance trajectories; the first column is the Google HD images; the second column is the Landsat true color composite images; the third column is the detected forest loss area in this study; the fourth column is the GFC forest loss area. The fifth column is the LandTrendr fitted disturbance trajectories based on NBR alone.

## 3. Results

### 3.1. Spatiotemporal Patterns in Forest Loss Area

The total forest loss area including both temporary and permanent (land use change) loss was $0.64 \times 10^4$ km$^2$ in Zhejiang Province during 1986–2019, with a mean annual loss rate of 188 km$^2$ (0.46%; Figures 5 and 6). This accounted for about 15.53% forest area in 1986, implying that 15.53% of forest area has been disturbed during the study period. The calculated permanent loss area was 5097 km$^2$, which accounts for about 79.63% of the total loss area, indicating that most of lost forest area in Zhejiang was due to land use change. The detected annual forest loss area showed no obvious change trend during 1986–1999, while a sharp decline from 2000 to 2003. Forest loss area showed an increasing trend from 2003 to 2008, and then kept no obvious change trend. Overall, the forest loss area showed a significant increasing trend (*slope* = 8.30 km$^2$/yr; $R^2$ = 0.29; *p* < 0.05) from 2003 to 2019. The highest loss area (379 km$^2$) occurred in 2017.

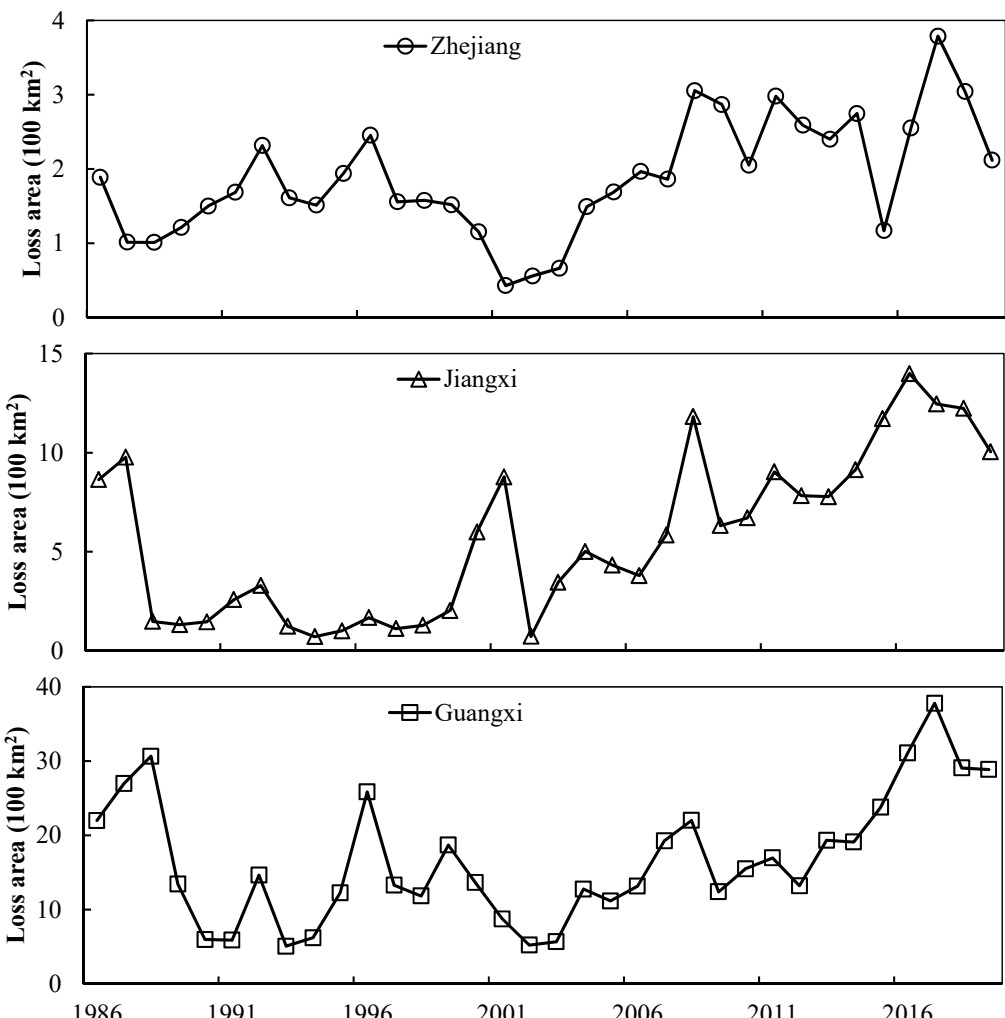

**Figure 5.** Interannual variations in detected forest loss area (km²) in Zhejiang, Jiangxi, and Guangxi Province during 1986–2019.

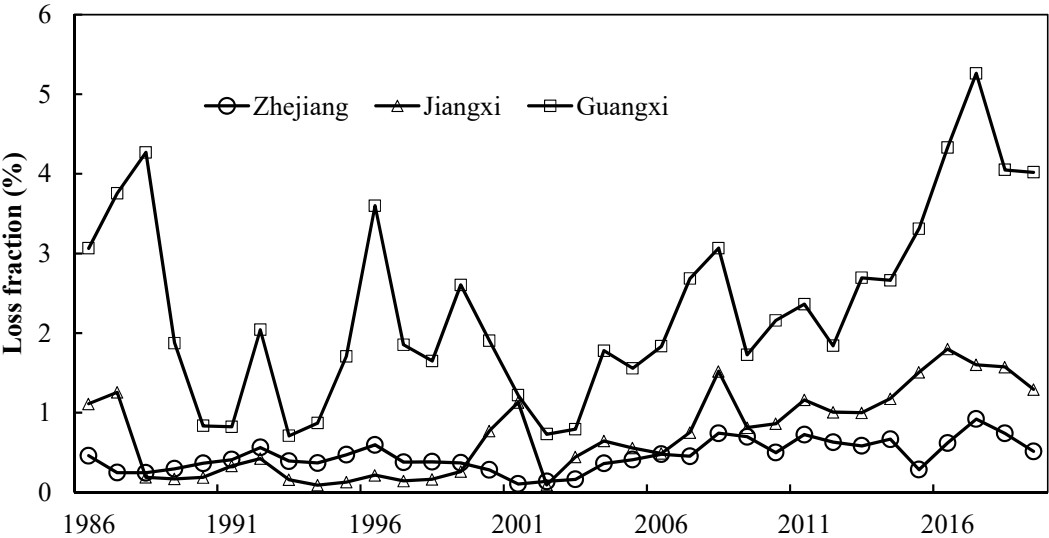

**Figure 6.** Interannual variations in detected forest loss fraction (%) relative to forest area in 1986 in Zhejiang, Jiangxi, and Guangxi Province during 1986–2019.

The total forest loss area was $1.94 \times 10^4$ km$^2$ in Jiangxi Province during 1986–2019, with a mean annual loss rate of 572 km$^2$ (0.74%). This accounted for about 25.02% forest area in 1986. The calculated permanent loss area was 5880 km$^2$, which accounts for about 30.24% of the total loss area, indicating that most of lost forest area in Jiangxi was due to temporary disturbance. The detected annual forest loss area showed a rapid decline from 1987 to 2003, while an obvious increasing trend was found from 2003 to 2016, and then kept a slight decline since 2017. Overall, the forest loss area showed a significant increasing trend (*slope* = 54.56 km$^2$/yr; $R^2$ = 0.69; $p$ < 0.01) from 2003 to 2019. The largest loss area (1399 km$^2$) occurred in 2016, and the forest loss area in 1987 (977 km$^2$), 2001 (877 km$^2$), and 2008 (1181 km$^2$) were also very large, indicating stronger disturbance in these years.

The total forest loss area was $5.72 \times 10^4$ km$^2$ in Guangxi Province during 1986–2019, with a mean annual loss rate of 1682 km$^2$ (2.34%). This accounted for about 79.62% forest area in 1986. The calculated permanent loss area was 4806 km$^2$, which accounts for about 8.40% of the total loss area, indicating that most of lost forest area in Guangxi was due to temporary disturbance. The detected annual forest loss area showed an increase from 1986 to 1988 and then a rapid decline from 1988 to 2003, with large interannual variation. An obvious increasing trend was found from 2003 to 2017, and then kept a slight decline since 2018. Overall, the forest loss area showed a significant increasing trend (*slope* = 139.70 km$^2$/yr; $R^2$ = 0.70; $p$ < 0.01) from 2003 to 2019. The largest loss area (3779 km$^2$) occurred in 2017, and the forest loss area in 1987 (2697 km$^2$), 1988 (3066 km$^2$), 1996 (877 km$^2$), and 2008 (2586 km$^2$) were also very large, indicating stronger disturbance in these years.

Totally, forest loss area was $8.30 \times 10^4$ km$^2$ in the Zhejiang, Jiangxi, and Guangxi Provinces during 1986–2019, accounting for 43.52% of total forest area in 1986 ($19.08 \times 10^4$ km$^2$). Compared with Jiangxi and Zhejiang, Guangxi Province, where is the current national timber base, has the largest forest area, loss area, and loss fraction in all years (Figure 5). The forest area in Jiangxi Province ($7.77 \times 10^4$ km$^2$) is significantly larger than Zhejiang ($4.12 \times 10^4$ km$^2$). Jiangxi Province has smaller loss fraction than Zhejiang Province during 1989–2000, while it was opposite during other years (Figure 6). The increasing trends in loss area during 2003–2019 ranked by Guangxi (139.70 km$^2$/yr) > Jiangxi (54.56 km$^2$/yr) > Zhejiang (8.32 km$^2$/yr). Although the loss area and change trends were different among three provinces, the interannual variations were quite similar. All provinces observed a significant increasing trend in forest loss area since 2003 and forest loss area was larger in 2008 and 2017.

At spatial scale, most forest loss areas were located in the forest edge or near the human settlements in all three provinces (Figures 7 and 8). In Zhejiang Province, larger forest loss areas were distributed in the northwest and southwest (Figure 8). At city level, Lishui City, located in the southwest, has the largest total loss area (2217 km$^2$) and loss fraction (20.38%) during 1986–2019. In Jiangxi Province, larger forest loss areas were distributed in the central west and south. At city level, the largest total loss area occurred in Ganzhou City (5438 km$^2$) and Jian City (3981 km$^2$); however, the largest loss fraction occurred in Xinyu City (30.32%). In Guangxi Province, forest loss areas were scattered across the entire province, with relatively smaller area in the central. At city level, the largest total loss area occurred in Baise City (8619 km$^2$) and Hechi City (7085 km$^2$); however, the largest loss fraction occurred in Yulin City (49.16%).

### 3.2. Spatiotemporal Patterns in Forest Gain Area

The total forest recovery area including both temporary (reforestation) and permanent (afforestation) gain was $2.54 \times 10^4$ km$^2$ in Zhejiang Province during 1986–2015, with a mean annual gain area of 848 km$^2$ (2.06%; Figure 9). This accounted for about 61.73% forest area in 1986, implying that 61.73% forest area has been gained during the study period. The calculated permanent gain (afforestation) area was $2.02 \times 10^4$ km$^2$, which accounted for about 79.55% of the total recovery area, indicating that most recovered forest area in Zhejiang was due to afforestation. The detected annual forest gain area showed a sharp decline from 1986 to 1991, while a larger interannual variation was observed during

1992–2003. The gained forest area showed a slight ($p$ = 0.69) increasing trend from 2003 to 2015. The highest gain area (2902 km$^2$) occurred in 1987.

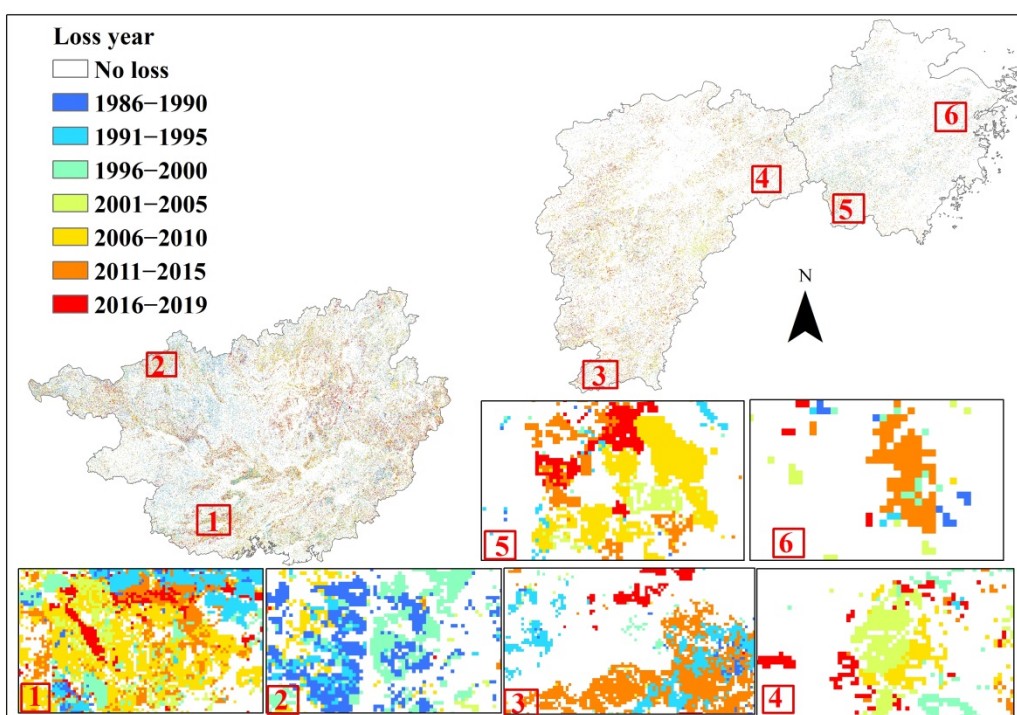

**Figure 7.** The spatial and temporal distribution of detected forest loss areas in Guangxi, Jiangxi, and Zhejiang Provinces during 1986–2019. Note: the six boxes at the bottom represent the magnified typical forest loss areas. Note: the numbers (1–6) denote the selected typical areas in the three provinces.

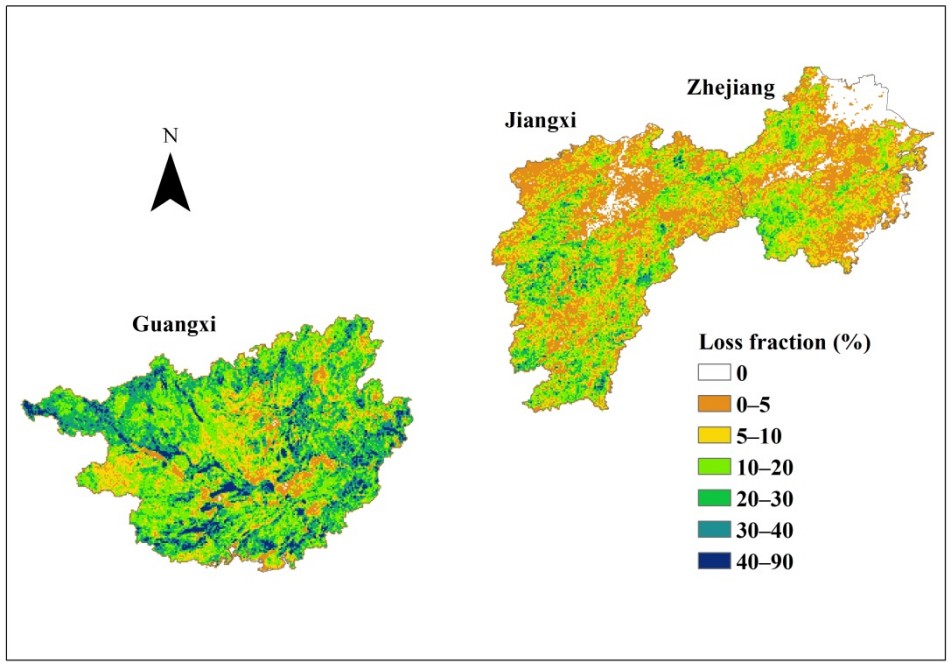

**Figure 8.** The aggregated (3 km × 3 km) total forest loss fractions in Guangxi, Jiangxi, and Zhejiang Provinces during 1986–2019.

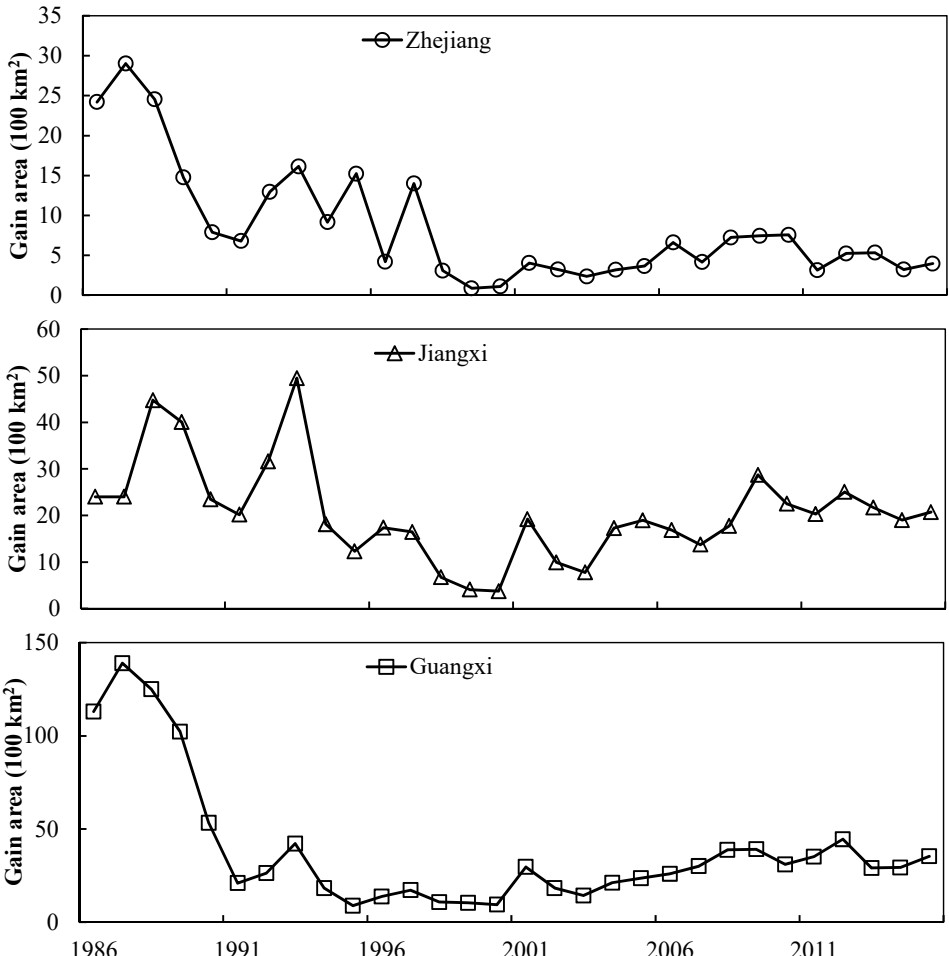

**Figure 9.** The interannual variations in forest gain area in the Guangxi, Jiangxi, and Zhejiang Provinces during 1986–2015.

The total forest gain area was $6.15 \times 10^4$ km$^2$ in Jiangxi Province during 1986–2015, with a mean annual gain area of 2052 km$^2$ (2.64%). This accounted for about 79.21% forest area in 1986. The calculated afforestation area was $3.61 \times 10^4$ km$^2$, which accounts for about 58.61% of the total gain area. The detected annual forest gain area showed a large variation during 1986–1995, but the mean annual gain area of this period was significantly larger than other years. Forest gain area showed an obvious increasing trend (18.24 km$^2$/yr; $R^2 = 0.54$; $p < 0.01$) from 2003 to 2012. The largest gain area occurred in 1988 (4474 km$^2$) and 1993 (4946 km$^2$).

The total forest gain area was $11.56 \times 10^4$ km$^2$ in Guangxi Province during 1986–2015, with a mean annual gain area of 3853 km$^2$ (5.36%). This accounted for about 160.89% of forest area in 1986, indicating recovered forest area was significantly larger than the original forest area. The calculated afforestation area was $9.25 \times 10^4$ km$^2$, which accounts for about 80.04% of the total gain area, indicating that most of recovered forest area was due to land use conversion to forests. The detected annual forest gain area showed a significant declining trend from 1986 to 1991 and then kept relatively stable during 1992–2002. An obvious increasing trend (38.41 km$^2$/yr; $R^2 = 0.80$; $p < 0.01$) was found from 2003 to 2012. The gain area decreased since 2013. The largest gain area occurred in 1987 ($1.39 \times 10^4$ km$^2$).

Totally, forest gain area was $20.25 \times 10^4$ km$^2$ in the Zhejiang, Jiangxi, and Guangxi Provinces during 1986–2015, accounting for 106.19% of total forest area in 1986. Compared with Jiangxi and Zhejiang, Guangxi Province had the largest gain area and fraction in most years, while Jiangxi had higher recovery fraction since 2003 as compared with Zhejiang. The increasing trends in loss fractions ranked by Guangxi > Jiangxi > Zhejiang during

the 2003–2012 period. The interannual variation patterns in gain area also showed some similarities. Forest gain area for all provinces had a decrease from 1986 to 1991 and an increasing trend during 2003–2012, and kept relatively stable during other years. The largest gain area showed in 1987 and 1988 for all provinces.

At spatial scale, the recovered forest areas generally scattered across the entire study region, with more gain areas at the fringes of forests (Figures 10 and 11). In Zhejiang Province, the largest gain area was mostly located at the southwest and northwest, with the largest gain area in Lishui City (5051 km$^2$) and the highest gain fraction in Jiaxing City (98.83%) during 1986–2015. In Jiangxi Province, the largest gain area was distributed in the southwest and central west, with the largest total gain area occurred in Ganzhou City ($1.74 \times 10^4$ km$^2$) and Jian City (9354 km$^2$); however, the largest gain fraction occurred in Nanchang City (73.69%). In Guangxi Province, the largest gain area was distributed in the northwest and the smaller gain area was scattered in the central. The largest total gain area occurred in Baise City ($2.06 \times 10^4$ km$^2$) and Hechi City ($1.77 \times 10^4$ km$^2$); however, the largest gain fraction occurred in Qinzhou City (108.78%).

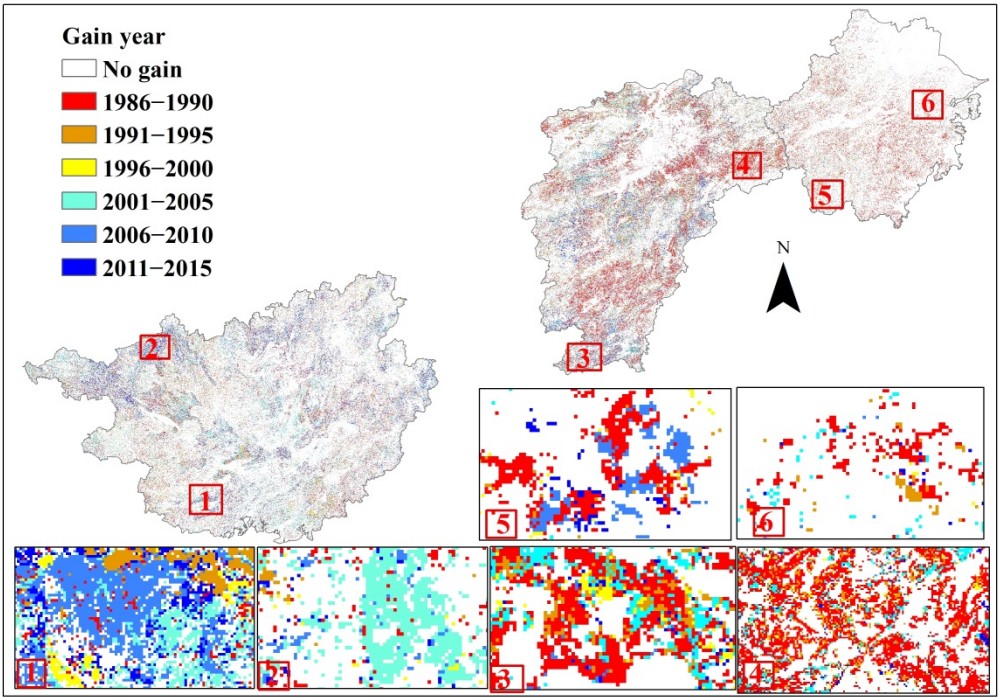

**Figure 10.** The spatial and temporal distribution of detected forest gain areas in the Guangxi, Jiangxi, and Zhejiang Provinces during 1986–2015. Note: the six boxes at the bottom represent the magnified typical forest gain areas. Note: the numbers (1–6) denote the selected typical areas in the three provinces.

### 3.3. Impacts of Forest Policies and Elevation on Forest Loss and Gain

During 1986–2019, some forestry policies and regulations have significantly affected the interannual variations and change trends of the forest loss (Figure 12). The "three determination" forestry policy (was also called the first collective forest tenure reform) in 1981 allowed the private parties to own and manage the collective forests, and the following forestry law in 1984 provided institutional support. This resulted in a gradual increase in forest losses since 1981 and it arrived at the highest loss area in 1986 and 1987. The central government realized the immaturity and big flaws in the "three determination" policy and then put forward the guidelines on enhancing the management of collective forest resource and prohibiting of the indiscriminate tree felling in 1987, which resulted in sudden decreases in forest loss area since 1986 for Zhejiang and Jiangxi and since 1987 for Guangxi. The implementation of the natural forest conservation program (NFCP) in 1998 slightly reduced forest loss area since 2000 and

resulted in the lowest forest loss area in 2002. The forest tenure reform policy trial in 2003 and full implementation in 2008, which allows private parties to own, manage and trade forest lands, resulted in greater forest loss in the three provinces, with more significant increasing trend in Guangxi. The full implementation the NFCP in 2010 and the protection policy for public welfare forest in 2013 slightly reduced the increasing trend in forest losses. The implementation for the reform of state-owned forest farms in 2015, which transform the profiting-making state-owned forest farms to a forest protection agency, resulted in a decline in forest loss after 2017. Although the key turning points and interannual variation patterns of forest loss area were similar among three provinces under the influences of similar national forest policies, the magnitudes of loss area and change trends were significantly different. This may be caused by the different implementation levels and local forest resource, climate and economy conditions. For example, all three provinces have implemented the Project for Fast-growing and High-yield Plantation in Key Areas. However, due to the suitable climate condition and more available non-forest land area, the wide planting of shorter rotation eucalyptus in Guangxi Province has resulted in larger forest loss area than other two provinces. Both Guangxi and Jiangxi Provinces were oriented and delineated as an agriculture-based province in the national economy system, thus the economy development more relies on the forestry output values than that in Zhejiang Province. This explained the higher forest loss area and greater change trends in forest loss in Guangxi and Jiangxi.

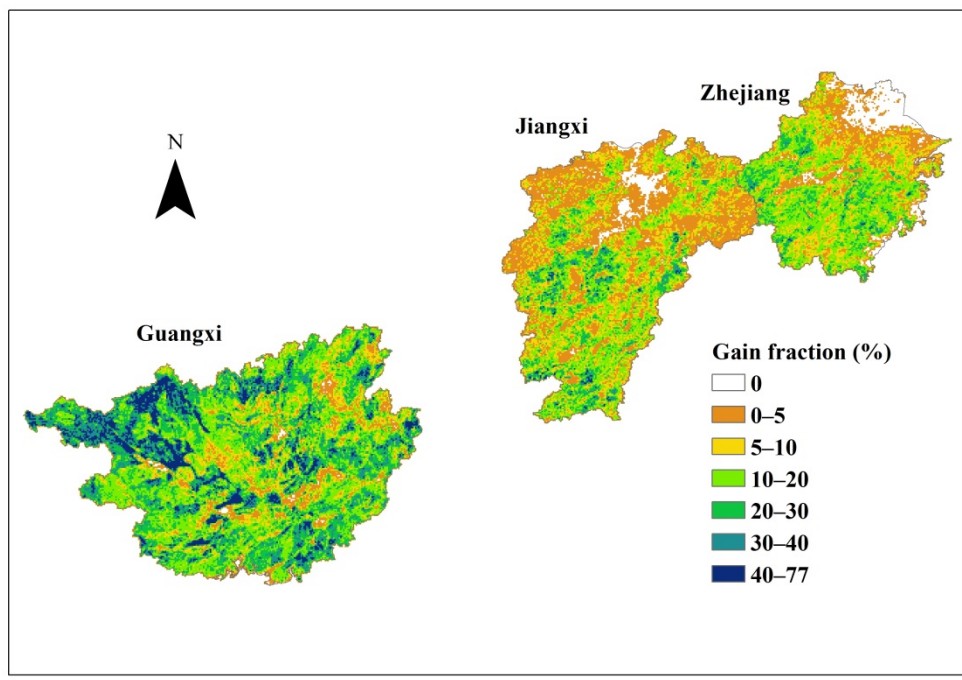

**Figure 11.** The aggregated (3 km × 3 km) total forest gain fractions in the Guangxi, Jiangxi, and Zhejiang Provinces during 1986–2015.

Similarly, most above forestry policies also affected forest gain patterns in the three provinces since most forest loss area recovered either through natural regeneration or reforestation. The "three-determination" policy in 1981 caused rapid increases in forest loss within several years (1982–1987), which resulted in subsequent large area forest gain during 1986–1990 (Figure 9). The legacy effect has been diminished and stabilized during 1991–2002. The implementations of GTGP policy in 2002 and the CFTR policy in 2003 had resulted in continuously increasing forest gain area since 2003. However, with almost no more non-forest land for afforestation and reduced forest loss due to forest protection policies, the forest gain area is anticipated to drop down significantly since 2017.

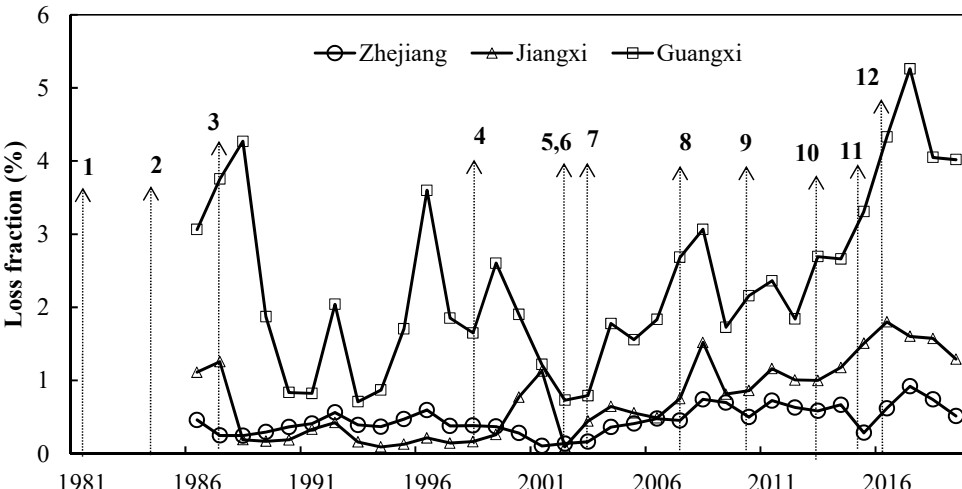

**Figure 12.** The timelines of the major forestry policies and interannual variations in forest loss area in the Zhejiang, Jiangxi, and Guangxi Provinces. 1: the three-determination policy (the first try for collective forest tenure reform) (1981); 2: the first Forestry Law of the People's Republic of China (1984); 3: policy for enhancing forest management and ban on indiscriminate tree felling (1987); 4: the trial of Natural Forest Conservation Program (NFCP) (1998); 5: Grain-to-Green (or Green for Grain) Policy (GTGP) (2002); 6: Project for Fast-growing and High-yield Plantation in Key Areas (FHPKA) (2002); 7: the trial of Collective Forest Tenure Reform (CFTR) (2003); 8: the full implementation of CFTR (2008); 9: the full implementation for NFCP (2010); 10: policy for public welfare forest classification and management (2013); 11: reform of the state-owned forest farms (RSFF) (2015); 12: the amendment of Forestry Law (CFTR, NFCP, and other policies were added into law) (2016).

## 4. Discussion

### 4.1. Comparisons with Other Data Sources

Our detected forest loss area was compared with the inventory data from NFI and Forestry Statistical Yearbook during the seven five-year periods (Figure 13). The inventory forest loss area was from four disturbance regimes including forest fire, insect, disease, and harvesting. The inventory data varied much more than our detected data. Our detected forest loss area was significantly lower than the inventory during 1989–1993, 1999–2003, 2004–2008, and 2009–2013 periods, while our detected forest loss area was higher during the rest periods. Our detected mean annual loss area (2452 km$^2$) was comparable to the inventory data (2673 km$^2$). From the inventory data, we found that forest loss due to insects accounted for a large fraction of total loss area. Our forest change detection algorithm can only effectively detect the areas with tree loss fraction greater than 70% at the grid size of 30 m × 30 m [18,35]. Forest pests and diseases generally cause partial tree losses and the impact area generally is smaller than 900 m$^2$, which resulted in failed detections for these areas and thus our detected forest loss area was less than the inventory data. We further compared our detected forest loss area with the GFC product [23], which is the first and most widely used forest loss and gain dataset. Compared with the GFC forest loss area (Figure 14), our detected forest loss area has the same interannual variation pattern during 2001–2019; however, our detected loss area was generally higher than the GFC data for most years. The GFC estimated a total loss of 4.77 × 10$^4$ km$^2$, while our detected loss area was 5.36 × 10$^4$ km$^2$, with 12% higher than the GFC data. Based on multiple forest change datasets, Li et al. [19] found that the total forest loss area from MODIS NBR, MODIS LC, and MODIS VCF products was 7206 km$^2$, 2.61 × 10$^4$ km$^2$ and 1.00 × 10$^4$ km$^2$, respectively, during 2001–2013 in the three provinces; while our detected total loss area was 2.82 × 10$^4$ km$^2$. Our data are close to the MODIS LC product but significantly higher than the loss area from other two products. The short-term forest loss cannot be reflected effectively based on the MODIS products.

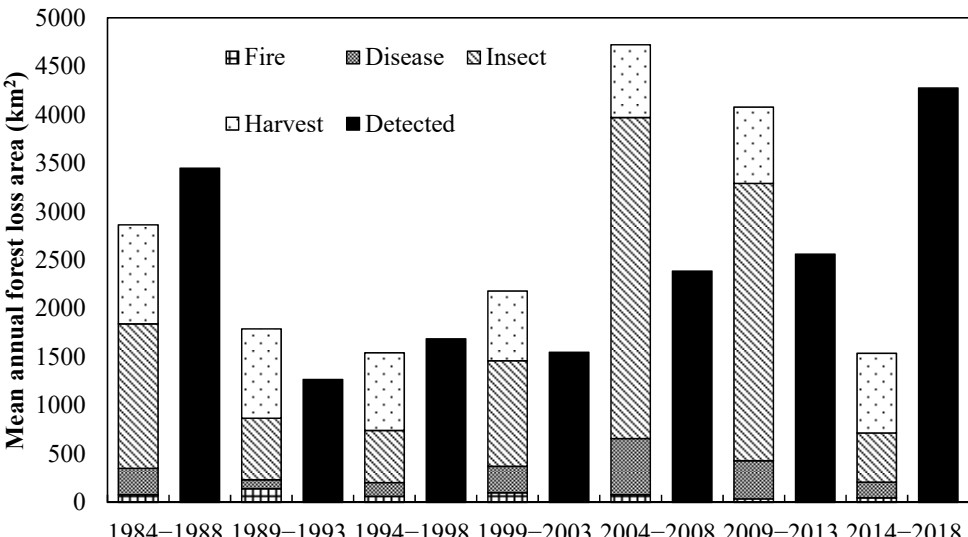

**Figure 13.** Comparison between our detected forest loss area and from NFI and Forestry Statistical Yearbook in each five year period during 1986–2018. Note: the loss area due to forest insect and disease is calculated by subtracting the prevention area from the total occurrence area.

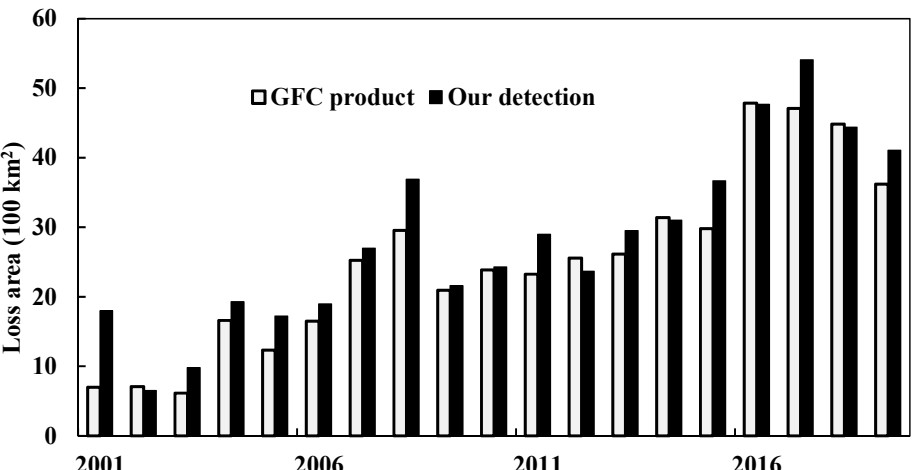

**Figure 14.** Comparisons of interannual variations in forest loss area (100 km$^2$) between our detection and the GFC product in the study region during 2001–2019.

We further compared our detected forest gain area with the NFI data (https://forest.ckcest.cn/sd/si/zgslzy.html; accessed on 15 March 2022). The 3rd and 9th NFI estimated forest area in the three provinces was $1.53 \times 10^5$ km$^2$ during 1984–1988 and $3.06 \times 10^5$ km$^2$ during 2014–2018, respectively. The net gain forest area was $1.53 \times 10^5$ km$^2$ during 1984–2018. Due to the change of forest definition since the 6th NFI (forest was defined by the land area covered by 20% trees before the 6th NFI and changed to 10% thereafter), the actual forest gain area should be slightly lower than $1.53 \times 10^5$ km$^2$. Our detected total forest gain area was $2.03 \times 10^5$ km$^2$, and the net gain area (the naturally recovered area was removed) was $1.49 \times 10^5$ km$^2$, which was very close to the NFI data.

The detected forest gain area was also compared with the annual planted forest area data from the Forestry Statistical Yearbook (Figure 15). From the comparison, our detected forest gain area was generally larger than the inventory planted forest area. This is because our detected forest gain area included both afforested area and naturally recovered area. The large forest loss area during the 1986–1988 period caused a large gain area due to naturally recovery during this period, which resulted in larger difference between our forest gain area and the inventory planted area. The interannual variation patterns between

two datasets were actually quite similar, but our data were generally a couple of years lagging behind the inventory data, which is reasonable because forest regrowth after planting need to take a couple of years to become greener and occupy he vacant land area and then the forest can be detected by satellites.

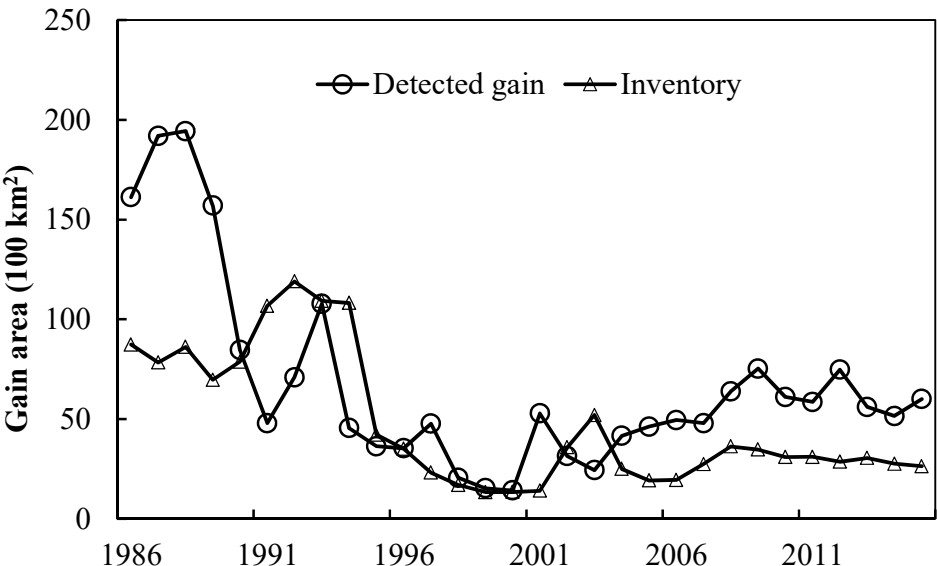

**Figure 15.** Comparisons of our estimated total forest gain area (100 km$^2$) with the NFI recorded planted forest area (https://forest.ckcest.cn/sd/si/zgslzy.html; accessed on 1 February 2022) in three provinces during 1986–2015.

Li et al. [19] also calculated the total forest gain area from MODIS NBR, MODIS LC, MODIS VCF, and GFC products: $0.79 \times 10^4$ km$^2$, $5.87 \times 10^5$ km$^2$, $2.12 \times 10^5$ km$^2$, and $0.70 \times 10^4$ km$^2$, respectively, during 2001–2013 in the three provinces; while our detected total gain area was $6.83 \times 10^5$ km$^2$. Our data is closer to the MODIS LC product but significantly higher than other three products. Song et al. [53] developed the global tree canopy change data and estimated that the total forest gain area was $5.18 \times 10^5$ km$^2$ during 1986–2015, which is slightly lower than our estimation. As implied from the NFI forest gain area, most of the previous datasets have underestimated the forest gain area in these three provinces, and may be also for the entire China. Some studies have also pointed out that GFC data lacked sufficient ground observation data to train the estimation method and verify the results, especially for China, where has experienced significant and complicated forest changes [16,19,43]. Our study obtained a large number of quadrat data for training the algorithm and validating the results, and thus gained a better detection result for both forest loss and gain in Zhejiang, Jiangxi, and Guangxi Provinces.

*4.2. Forestry Policies and Forest Loss and Gain Area*

The classified forest dynamics based on remote sensing images have been widely applied to study the effects of forestry policies on forest structure and function. For example, Velasco-Aceves [54] applied Landsat images assessed the outcome of the first land-use zoning policy in terms of forest loss and fragmentation in Formosa, Argentina. West et al. [55] reviewed the conservation instruments and strategies promoted under the PPCDAm umbrella and estimated their impacts on deforestation based on rigorous and counterfactual evaluations. Using Landsat imagery, Appiah et al. [56] used a RF machine learning algorithm analyzed the amount and pattern of land changes in the Tano-Offin Forest Reserve of Ghana between 1987 and 2017. In our study, we found that the "three determination" policy put forward in 1981 has greatly increased forest loss area during 1986–1988. This policy was put forward to determine forest use rights, delineate self-retained mountainous land for farmers and standardized both family forest management and the household responsibility

system within the collectives [5]. Many previous studies have also found the resultant increased forest loss due to this policy [4,7,13,57]. For example, Liu et al. [5] indicated that as much as 13.5% of forest area and 14.9% of forest volume was removed during 1980–1990 in southern China. Their study further attributed the increased forest harvest to the "three determination" policy. The NFI data also indicated harvested forest area significantly increased during 1984–1989 in all three provinces [5]. Yin and Newman [57] also observed the increased forest loss area and attributed to this policy. Hyde [8] also indicated that forest harvesting area increased by 10.4% and standing volume decreased by 9.6% during 1978–1989, which was also confirmed by [58]. The followed-up forest management and protection policy raised in the "Guidelines on Enhancing the Management of Collective Forest Resources in the South and Prohibiting of the Indiscriminate Tree Felling" in 1987 reversed the uncontrolled logging due to "three determination" policy and called for an end to this policy [5]. Our results also proved that this policy can effectively reduce forest loss area in the Zhejiang, Jiangxi, and Guangxi Provinces. Our results indicated that the natural forest conservation project (NFCP) in 1998 did not affect the forest gain area but it effectively reduced forest loss area during 2000–2002, which has also been indicated in many previous studies [3,7,12,59,60]. Van Den Hoek et al. [61] applied Landsat imagery-derived Tasseled Cap variables and a decision tree classifier to map short- and long-term forest cover change and assessed the effectiveness of NFCP and GTGP policies on forest cover in Yunnan Province. Their study also found that these two policies resulted in 73% reduction of forest loss and 100% increase in forest gain in Yunnan Province, neighboring Guangxi Province.

A series of policies and regulations have been put forward in 2002 and 2003, including the GTGP, FHPKA, and CFTR. These policies coincided and positively contributed to the increases in both forest loss and gain area, which has been proved by a bunch of studies [3,4,8,62]. The FHPKA and GTGP are two of the China's biggest afforestation projects. GTGP has less impacts than FHPKA on the forest dynamics in southern China [61]. The FHPKA project targets to provide wood products to the increasing demand of wood market in China and in the meantime to protect natural forests from logging [63]. This project was applied in 18 provinces, mostly located in the south. The major tree species include Eucalyptus, poplar, improved Chinese fir, and bamboo. This policy has promoted both increases in forest loss and gain area, which is one of the major reasons for the increasing forest gain area and loss area during 2003–2019. Research showed about 3570 km$^2$ forest area was planted due to this project during 2002–2009 [64]. This project imposed more impacts on Guangxi Province [19,65], which is also proved from the larger gain and loss area and faster increasing trend since 2003. As a result, Guangxi Province became the largest timber supplier in China. Since 2003, the second CFTR proved to be a suitable policy for forest resource management in China; therefore, it was fully implemented in 2008 and most provinces have completed the reform tasks at present [4,13]. This policy has successfully put China's forests in the trading market, increased farmers' income, and in the meanwhile, has reduced illegal forest logging and protected the natural forests [13]. At present, the effects of the CFTR on forest dynamics were mostly estimated based on the NRI inventory data or the Forestry Statistical Yearbook statistical data, which are generally discontinuous and coarse. The spatiotemporally-explicit and more accurate assessments are needed. The protection policy for public welfare forests in 2013, Reform of State-owned Forest Farms (RSFF) in 2015, and the amendment of forestry law in 2016 have successfully promoted the forest protection in China, resulting in declining trend in forest loss area as seen from our study results. The forestry law amended in 2019 added stricter forest protection terms and the "two-mountains" (lucid waters and lush mountains are invaluable assets) and ecological civilization guiding ideology. Therefore, the declining trend of forest loss area may continue in the near future and then stabilize. In another aspect, there is less available land for afforestation in China, especially in the south. In addition, our results found most of the past forest gain area in the three provinces was due to afforestation (land

conversion to forest). Therefore, a declining trend of forest gain area will continue and forest gain area will be more from reforestation on the logged forest land in the near future.

### 4.3. Forestry Economy and Forest Loss

The economy policy can also indirectly affect forest loss area and explain the different change trends [58,62]. Based on the NFI data, Liu and Xia [62] found that the socio-economic policies can significantly interact with forestry policy to influence forest loss and gain dynamics in China. Based on the forestry output value data from the NFI, we also conducted a correlation between forestry output value and forest loss area during 1990–2019. The results indicated that forestry output value significantly correlated with forest loss area in the Jiangxi ($R^2 = 0.59$; $p < 0.01$), Guangxi ($R^2 = 0.76$; $p < 0.01$), and Zhejiang ($R^2 = 0.34$; $p < 0.01$) Provinces (Figure 16). Inferred from the slopes, the impact levels of economy can be ranked as Guangxi > Jiangxi > Zhejiang. Guangxi Province is now the timber base for China, which was shifted from northeastern China [65]. Guangxi only accounts for 5% of the national forest land area, but contributed to about 50% of the national total wood products in 2020. Due to perfect climate conditions and high profit return, the "Project for Fast-growing and High-yield Plantation in Key Areas" has promoted a rapid increase in eucalyptus forests in Guangxi Province and thus wood products, which further stimulated the forestry economy growth in this province. In contrast, the forestry income in Jiangxi Province was mainly from large plantation area for *Cunninghamia lanceolata* (Chinese fir), *Pinus massoniana* (Masson pine), and *Pinus elliottii* (Slash pine). These species have longer rotation and thus the forestry output value and its increasing trend were lower than that in Guangxi. Zhejiang Province is the earliest province that implemented the policy for NFCP and for protection of ecological public welfare forests. Over 10% of the forestry output value was from the bamboo industry [8]. The bamboo can recover soon and thus bamboo cutting is not accounted in forest loss area.

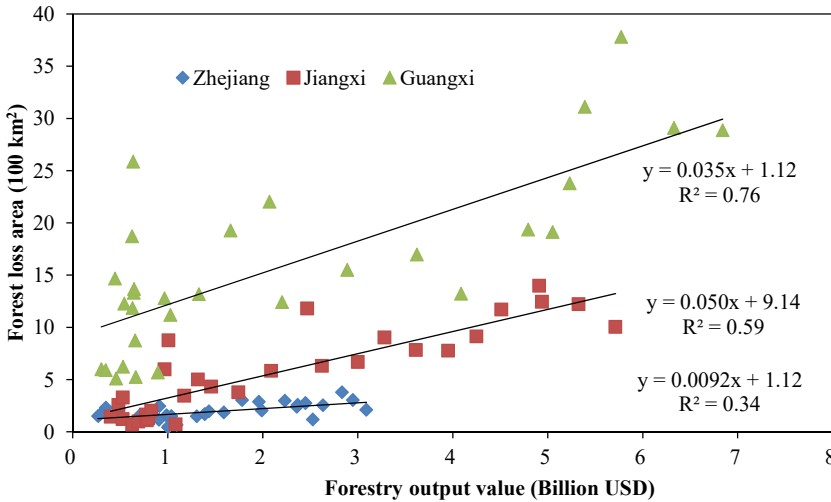

**Figure 16.** The regressions between forestry economy (forestry output value; Billion USD) and our detected forest loss area in Zhejiang, Jiangxi and Guangxi Provinces.

### 4.4. Uncertainty and Outlook

Although we have evaluated the detected results against the data from sampling plots, inventory and other remote sensing products, there are still some uncertainties in the detected forest loss and gain area. The success of LandTrendr algorithm relies heavily on the available image numbers and quality. As shown in Figure 2, the cloud-free Landsat images scenes were not sufficient during 1986–2000, much less than that after 2001. Some of the cloudy pixels were substituted by the pixels in the neighboring years before 2000, which may result in underestimation of detected forest loss and gain area. In addition, the resolution of Landsat image is relatively coarse, and thus there are many pixels mixed with

different forest status. Our current approach can only detect the standing-replace (>70% trees are altered within a grid cell) forest loss and gain events. Thus, the small scale forest loss and gain caused by fractional disturbance and recovery events cannot be detected. As we shown above, large area of forest in the three provinces suffered from infections of forest insect and disease, and under most cases these events can only cause scattered tree losses. Similarly, the small-scale tree planting along the riverside and roadside cannot be effectively detected. Therefore, this may also result in some underestimations to the forest loss and gain area. Furthermore, our approach exists a lagging for the monitoring of forest recovery. From the comparisons with inventory planted forest area, we deduced that the lagging is about two to three years (Figure 15). This is because vegetation recovery generally takes several years to occupy the vacant space and our approach is more sensitive to the stand-replacing events; therefore, the detected forest gain years generally deviate within 0–5 years depending on the study locations.

The single factor could result in larger bias for forest loss and gain detection considering the image quality issues [35]. Through considering more variables except for the normally used individual indices such as IFZ, NBR, TCW, EVI, and NDVI, machine or deep learning methods (such as RF, ANN, and RCNN) have been increasingly applied in forest loss and gain detection during the recent years, and these methods have been proved to more comprehensively and accurately reflect both slow and abrupt forest loss and gain events [18,33,35–38]. Our study also proved that this integration method can be used to detect provincial scale forest loss and gain in China, where has more complex terrains, vegetation types, and disturbance events. The major limitation for this approach is the heavy reliance on the accuracy and numbers of sample plot data for training the machine-learning algorithms. In the next step, we will further collect more sample plot data and apply this integrated approach to develop an accurate, long-term, continuous, and high spatiotemporal resolution national scale forest loss and gain dataset, similar to the datasets for the United States [22,25,27,66]. In addition, we will detect the fractional forest loss and gain and further differentiate the forest disturbance regimes in the future. As inferred from many cutting-edge studies, the forest disturbance regimes and fractional disturbance events can be successfully detected using the 30 m Landsat images [33,66–69].

## 5. Conclusions

Based on Landsat time series images and the GEE platform, this study detected forest loss and gain area during 1986–2019 in the Zhejiang, Jiangxi, and Guangxi Provinces in the subtropical China by using integrated LandTrendr change detection algorithms and RF classifier. The detected forest loss and gain area and their variation patterns were evaluated against sampling plot data, other data products, and inventory data. The evaluations indicated our approach can provide accurate estimations on the spatiotemporal patterns of forest loss and gain area. The results indicated that a large portion (43.52%) of forests in the three provinces had been disturbed and in the meanwhile forest area was doubled during the study period. Three provinces displayed contrasting patterns in forest loss and gain area, interannual variation patterns and change trends. The different footprints of the similar forestry policies and regulations on three provinces can explain most of the similarities and dissimilarities of forest gain and loss area in three provinces. Our study also anticipated a declining forest loss and gain area in future in these three provinces. Although some uncertainties exist, our study can contribute to a better understanding of forest dynamics in the subtropical China through providing a 34 years' long annual forest loss and gain dataset. The further assessments of the effectiveness of major forestry policies and regulations on forest loss and gain can also contribute to more effective implementation of these forest management policies, forest-based climate change mitigation policies and national carbon balance accounting [70].

**Author Contributions:** Conceptualization, J.S. and G.C.; methodology, J.S. and G.C.; validation, J.S., J.H., S.H. and J.M.; formal analysis, J.S. and J.H.; data curation, J.S., S.H. and J.M.; writing—original draft preparation, J.S.; writing—review and editing, G.C., J.H., S.H. and J.M.; visualization, J.S. and J.H.; supervision, G.C.; project administration, G.C.; funding acquisition, G.C. All authors have read and agreed to the published version of the manuscript.

**Funding:** This research was funded by the Natural Science Foundation of Zhejiang Province (Grant number LY20C030001), Guangxi Key Research and Development Program (Grant number AB21220057), Research Funds of the Guangxi Key Laboratory of Landscape Resources Conservation and Sustainable Utilization in Lijiang River Basin, Guangxi Normal University (Grant number LRCSU21K0101), the Scientific Research Foundation of Zhejiang A&F University (Grant number 2034020080), and the Overseas Expertise Introduction Project for Discipline Innovation (111 Project; Grant number D18008).

**Data Availability Statement:** All data generated or analyzed during this study are included in this article.

**Acknowledgments:** We acknowledge Shen, Zhenming from the Bureau of Agriculture and Rural Affairs of Lin'An District and Jiao, Hongbo from the Forestry Industry Development Administration of Xinyu for providing some sampling plot data.

**Conflicts of Interest:** The authors declare no conflict of interest.

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
