# Peer review of "Contrasting Forest Loss and Gain Patterns in Subtropical China Detected Using an Integrated LandTrendr and Machine-Learning Method"

_remotesensing, doi:10.3390/rs14133238_

Round 1
Reviewer 1 Report
I review this manuscript “Contrasting forest loss and gain patterns in the subtropical China detected using an integrated LandTrendr and machine learning method” and found that it is interesting in remote sensing fields. Whole text is easy to read and the flow is easy to follow. Moreover, I have some special comments as follows.
1. Title and Abstract are suitable for this paper.
2. In introduction chapter, I suggested that authors might emphasize the advantages of integrated LandTrendr and machine-learning in this fields
3. The framework of Materials and Methods is easy to understand.
4. Line 468, I suggest using “3. Results” to replace “3. Results and analysis”.
5. Year in Figures 7 and 10 using comma is not suitable. All years should be improved!
6. I found that many Figures were shown in Discussion chapter. If they were cited from references, they should be added reference in Figure titles.
7. I suggested that machine learning method should have more discussion in “4.4. Uncertainty and outlook” section (Line 967).
8. Conclusion is not suitable for citation. It should be based on the results of this study.
Overall, I am pleased to recommend this manuscript for publication in the remote sensing.

Reviewer 2 Report
This study evaluates forest loss and recovery patterns in the subtropical forests of China using the LandTrendr change detection algorithm with a Random Forest (RF) classifier. They mapped forest disturbance and recovery using the second classification proposed by Cohen et al. (2018). They found annual forest changes, which corresponded with forest policy and regulation.
Overall, the approach used in this manuscript is interesting because there are limited studies that used machine learning approaches for change detection using LandTrendr. However, I found many deficits in the introduction, preprocessing, and accuracy assessments in this manuscript. In particular, the contents of this manuscript are too similar with those of Hua et al. (2021). There is little contribution to the remote sensing community and novelty, in my view.
Major Comments:
1. The manuscript is too similar with Hua et al. (2021). The authors seemed to use almost the same methods for change detection, mapping, and accuracy assessment in different (but include the same) provinces. The flow and appearance of some figures are quite similar. The comparison with the existing dataset is also very similar. Although I understand that the objective of this manuscript is different from Hua et al. (2021), the influences of policy and regulation on forest changes are already discussed in Hua et al. (2021). I do not think the current manuscript can add any new insights to the remote sensing community.
2. Accuracy assessments should follow good practice recommendations (e.g., Olofsson et al. 2014). In particular, the pixel counting error matrix is usually biased. The authors should be based on an unbiased estimator. In addition, the procedure for collecting reference samples is not clear. Did the authors implement random sampling? The detailed description is required.
3. Many important references were not cited in the introduction and methods. For example, the authors failed to cite the GEE version of LandTrendr (Kennedy et al. 2018). There are several studies that used LandTrendr with RF for change detection (Nguyen et al. 2018; Bell et al. 2021; Shimizu and Saito 2021). There is no citation for GEE (Gorelick et al. 2017). There is no citation for BFAST, VCT, and CCDC.
4. Some citations are wrong or not appropriate. Several publications are duplicated in the reference list. The authors should check these references.
5. There are many abbreviations that are not mentioned. Some are not correct. The authors should check and provide what they are standing for.
The following are minor comments, but I cannot raise all minor concerns because there are too many deficits.
L22: The authors should provide the definition of forest loss and gain, and explain how they are different from forest disturbance and recovery in the main text.
L99: The citation for Cohen et al. (1998) is not suitable. Please check.
L119: Several citations can be added to the accuracy of Hansen GFC and local disturbance map (e.g., Galiatsatos et al. 2020; Bos et al. 2019; Jutras-Perreault et al. 2021).
L230: There is no citation for atmospheric correction of Landsat 8 OLI (Vermote et al. 2016). Maesk et al. (2006) is usually cited for LEDAPS, not Vermote et al. (1997).
L231: As far as I understand, preprocessing steps are not the parts of the LandTrendr algorithm. Please check.
L372: Kennedy et al. (2007) is not suitable here. Please check.
References:
Bell, D.M., Acker, S.A., Gregory, M.J., Davis, R.J., Garcia, B.A., 2021. Quantifying regional trends in large live tree and snag availability in support of forest management. For. Ecol. Manage. 479, 118554. https://doi.org/10.1016/j.foreco.2020.118554
Vermote, E., Justice, C., Claverie, M., Franch, B., 2016. Preliminary analysis of the performance of the Landsat 8/OLI land surface reflectance product. Remote Sens. Environ. 185, 46–56. https://doi.org/10.1016/j.rse.2016.04.008
Hua, J., Chen, G., Yu, L., Ye, Q., Jiao, H., Luo, X., 2021. Improved Mapping of Long-Term Forest Disturbance and Recovery Dynamics in the Subtropical China Using All Available Landsat Time-Series Imagery on Google Earth Engine Platform. IEEE J. Sel. Top. Appl. Earth Obs. Remote Sens. 14, 2754–2768. https://doi.org/10.1109/jstars.2021.3058421
Kennedy, R.E., Yang, Z., Gorelick, N., Braaten, J., Cavalcante, L., Cohen, W.B., Healey, S., 2018. Implementation of the LandTrendr Algorithm on Google Earth Engine. Remote Sens. 10, 691.
Olofsson, P., Foody, G.M., Herold, M., Stehman, S. V, Woodcock, C.E., Wulder, M.A., 2014. Good practices for estimating area and assessing accuracy of land change. Remote Sens. Environ. 148, 42–57. https://doi.org/10.1016/j.rse.2014.02.015
Shimizu, K., Saito, H., 2021. Country-wide mapping of harvest areas and post-harvest forest recovery using Landsat time series data in Japan. Int. J. Appl. Earth Obs. Geoinf. 104, 102555. https://doi.org/10.1016/J.JAG.2021.102555
Nguyen, T.H., Jones, S.D., Soto-Berelov, M., Haywood, A., Hislop, S., 2018. A spatial and temporal analysis of forest dynamics using Landsat time-series. Remote Sens. Environ. 217, 461–475. https://doi.org/10.1016/J.RSE.2018.08.028
Gorelick, N., Hancher, M., Dixon, M., Ilyushchenko, S., Thau, D., Moore, R., 2017. Google Earth Engine: Planetary-scale geospatial analysis for everyone. Remote Sens. Environ. 202, 18–27. https://doi.org/10.1016/j.rse.2017.06.031
Bos, A.B., De Sy, V., Duchelle, A.E., Herold, M., Martius, C., Tsendbazar, N.-E., 2019. Global data and tools for local forest cover loss and REDD+ performance assessment: Accuracy, uncertainty, complementarity and impact. Int. J. Appl. Earth Obs. Geoinf. 80, 295–311. https://doi.org/10.1016/J.JAG.2019.04.004
Galiatsatos, N., Donoghue, D.N.M., Watt, P., Bholanath, P., Pickering, J., Hansen, M.C., Mahmood, A.R.J., 2020. An Assessment of Global Forest Change Datasets for National Forest Monitoring and Reporting. Remote Sens. 12, 1790. https://doi.org/10.3390/RS12111790
Jutras-Perreault, M.-C., Gobakken, T., Ørka, H.O., 2021. Comparison of two algorithms for estimating stand-level changes and change indicators in a boreal forest in Norway. Int. J. Appl. Earth Obs. Geoinf. 98, 102316. https://doi.org/10.1016/j.jag.2021.102316
Masek, J.G., Vermote, E.F., Saleous, N.E., Wolfe, R., Hall, F.G., Huemmrich, K.F., Gao, F., Kutler, J., Lim, T.K., 2006. A Landsat surface reflectance dataset for North America, 1990-2000. IEEE Geosci. Remote Sens. Lett. 3, 68–72. https://doi.org/10.1109/LGRS.2005.857030
Reviewer 3 Report
The paper “Contrasting forest loss and gain patterns in the subtropical China detected using an integrated LandTrendr and machine-learning method” identified the forest loss and gain over three subtropical provinces in China during 1986-2019 from Landsat images. It combined LandTrendr and Random forest method to classify forest loss/gain/unchanged over time and space, and analyzed the impacts of policies and environment on forest resources changes. It is a good fit for the journal. The method is solid and the analysis is robust at good quality. Some minor comments are listed here:
1. Line 228:Please provide some evidence to show the quality of the conversion among LT missions. It could have significant impact on the time series analysis results, if it was not done correctly.
2. Table 2 and related analysis and discussion: many policies are listed but not well discussed. It is better to focus on the main policies that made substantial impacts on forest resources. Drawn a strong conclusion on what and how each polices are influencing the forest spatially and temporally.
3. Section 2.3.3 Is the secondary classification RF done for each year separately? How was the outputs from LandTrendr included in the classification? Need more explanation or a detailed example.
4. Figure 10: Suggest using a continuous color ranges (e.g. red to green). Easier to see the pattern of gain area in earlier and later years.
